# Nitrogen oxides and ozone fluxes from an oilseed-rape management cycle: the influence of cattle slurry application

Raffaella M. Vuolo[1], Benjamin Loubet[1*], Nicolas Mascher[1], Jean-Christophe Gueudet[1], Brigitte Durand[1], Patricia Laville[1], Olivier Zurfluh[1], Raluca Ciuraru[1], Patrick Stella[2] and Ivonne Trebs[3]

1 UMR ECOSYS, INRA, AgroParisTech, Université Paris-Saclay, 78850, Thiverval-Grignon, France
2 UMR SADAPT, AgroParisTech, INRA, Université Paris-Saclay, 16 rue Claude Bernard, 75231 Paris, France
3 Luxembourg Institute of Science and Technology (LIST), Environmental Research and Innovation (ERIN), 41, rue du Brill, L-4422 Belvaux, Luxembourg

* Corresponding author: Benjamin.Loubet@inra.fr

**Abstract.** This study reports NO, $NO_2$ and $O_3$ mixing ratios and flux measurements using the eddy-covariance method during a 7 month period over an oilseed rape field, spanning an organic and a mineral fertilisation event. Cumulated NO emissions during the whole period were in agreement with previous studies and showed quite low emissions of 0.26 kg N ha$^{-1}$ with an emission factor of 0.27%, estimated as the ratio between total N emitted in form of NO and total N input. The NO emissions were higher following organic fertilisation in August due to conditions favouring nitrification (soil water content around 20% and high temperatures), while mineral fertilisation in February did not result in high emissions. The ozone ($O_3$) deposition velocity increased significantly after organic fertilisation. The analysis of the chemical and turbulent transport times showed that reactions between NO, $NO_2$ and $O_3$ below the measurement height occurred constantly throughout the 7 month period. Following organic fertilisation, the NO ground fluxes were 30% larger than the NO fluxes at the measurement height (3.2 m), while the $NO_2$ fluxes switched from deposition to uptake during certain periods, being negative at the surface and positive at the measurement height. This phenomenon of "apparent $NO_2$ emissions" appears to be significant during strong NO emissions and high $O_3$ ambient mixing ratios, even on a bare soil during August.

**Keywords:** eddy covariance, chemical reaction, transport time, oilseed rape, NO, $O_3$, $NO_2$

## 1. Introduction

Agricultural soils represent an important source of atmospheric nitric oxide (NO), especially in highly fertilized regions (Oikawa et al., 2015). Global estimates of total $NO_x$ (NO + $NO_2$) emissions from soils range between 4 and 21 Tg N yr$^{-1}$, which represents between 10% and 15% of the global $NO_x$ budget (Davidson and Kingerlee, 1997; Houghton et al., 2001; Yienger and Levy, 1995). $NO_x$ inventories are subject to error in magnitude and especially in spatial distributions (Martin et al., 2003), which can be constrained by satellite observations and range around 30% at the global scale (Toenges-Schuller et al., 2006). $NO_x$ emissions are of considerable interest also for atmospheric photochemistry, and acting as ozone ($O_3$) precursors, they indirectly have an impact on climate. $O_3$ is indeed an important greenhouse gas, contributing 25% of the anthropogenic net radiative forcing (Forster et al., 2007).

NO$_x$, and especially NO$_2$, are toxic gases for humans, exposure to which increases risks of various respiratory diseases. The World Health Organization gives guidelines for NO$_2$ exposure limits, both annual means (40 µg m$^{-3}$) and 1-hour mean (200 µg m$^{-3}$) (Prüss-Üstün et al., 2016). For O$_3$, only a short-term threshold is given (100 µg m$^{-3}$ for the 8-hour mean) because there are fewer studies on long-term exposure. These thresholds are established both in epidemiological and toxicological studies on humans and animals. Similarly, nitrogen deposition leads to serious adverse effects on vegetation (eutrophication, biodiversity erosion and acidification being the most serious ones), while O$_3$ has a direct adverse effect on plant health through oxidation of photosynthesis pathways and direct tissue destruction above large thresholds. For nitrogen, the concept of critical load has been developed which gives the amount of nitrogen deposition above which an ecosystem is impacted. These critical loads range from 5 kg N ha$^{-1}$ yr$^{-1}$ for sensitive habitats to 20 kg N ha$^{-1}$ yr$^{-1}$ for less sensitive ones (APIS, 2016). For these reasons, national and international authorities regulate atmospheric levels of these pollutants.

NO$_x$ emissions from soils are primarily the by-products of nitrification and denitrification processes and the chemical decomposition of HNO$_2$ (Laville et al., 2005; Meixner, 1997; Remde et al., 1989). Many authors emphasize that for most agricultural soils, nitrification is the dominant process of NO emissions (Bollmann et al., 1999; Dunfield and Knowles, 1999; Godde and Conrad, 2000). Organic and mineral fertilizers, rich in ammonium, increase NO emissions both by stimulating NO production by nitrification and by decreasing NO consumption.

There is a significant knowledge gap in understanding NO$_x$ exchange between agricultural ecosystems and the atmosphere, partly due to a lack of direct measurements over long periods. NO emissions by soils can either be measured by dynamic chambers (Breuninger et al., 2012; Laville et al., 2009; Laville et al., 2011; Pape et al., 2009), aerodynamic gradient (Kramm et al., 1991), or eddy covariance methods (Rummel et al., 2002; Stella et al., 2013a). Each method has its drawbacks and challenges: the dynamic chamber method may change the surface exchange parameters (Pape et al., 2009), and modify the fluxes due to fast reactions between the triad O$_3$-NO-NO$_2$, but thoroughly designed Teflon chambers can overcome this problem (Skiba et al., 2009). The aerodynamic gradient method (AGM) is a well-established method applicable to water-soluble compounds such as NH$_3$ (Milford et al., 2009), but has several biases of which flux divergence due to chemical reaction is the most limiting for NO-NO$_2$-O$_3$ (Duyzer et al., 1995; Kramm et al., 1991; Loubet et al., 2013). Non-stationarity and integration time are also limiting (Lenschow et al., 1994; Stella et al., 2012). The eddy covariance method is adapted for measuring NO fluxes. It is however also vulnerable to flux divergence issues due to NO-NO$_2$-O$_3$ chemical reactions. It is therefore essential to measure the fluxes and mixing ratios of the three compounds together.

The eddy covariance (EC) method is the state of the art flux measurement method for energy and CO$_2$ fluxes (Baldocchi, 2003), and due to the development of new analysers such as fast chemiluminescence, quantum cascade lasers absorption spectroscopy, or proton time of flight mass spectrometers (Ammann et al., 2012; Brodeur et al., 2009; Ferrara et al., 2012; Li et al., 2013; Muller et al., 2010b; Park et al., 2014; Peltola et al., 2014; Sintermann et al., 2011; Stella et al., 2013a; Wolfe et al., 2009) it can currently be applied to several other trace gases. The main advantage of the EC method is that it is a "direct" measurement of the flux at a given height, which depends on fewer assumptions than the AGM, namely the Reynolds averaging and ergodicity hypothesis requiring that *"the averaging time must be much larger than the time scales of variation of the air*

*velocity"* (Corrsin, 1975, see also Kaimal and Finnigan, 1994). This method has been successfully applied for measuring NO fluxes in a limited number of studies (Eugster and Hesterberg, 1996; Lee et al., 2015; Marr et al., 2013; Min et al., 2014; Rummel et al., 2002; Stella et al., 2013a). The main difficulties of EC measurements are the losses that appear at high frequencies due to adsorption of the gas to the tubing system, which depends also on the size of the absorption cell (Eugster and Senn, 1995) and differential advection caused by the radial variation of the mean velocity and simultaneous radial diffusion of the sample gas (Lenschow and Raupach, 1991). Moreover, since $NO_2$-to-NO photolytic converters typically applied in combination with chemiluminescence analysers have a conversion efficiency below 100%, measuring both NO and $NO_2$ with such a converter remains a challenge that requires the use of two analysers simultaneously (Lee et al., 2015).

Due to these limitations, simultaneous measurements of NO, $NO_2$ and $O_3$ fluxes by eddy covariance have hence seldom been made. To our knowledge, only a few studies report such measurements and none over an arable crop. There is therefore a gap in knowledge as to whether the reactions between NO, $NO_2$ and $O_3$ significantly influence the fluxes above crops and how nitrogen application modifies these fluxes and their interactions. Eugster and Senn (1995) report $NO_2$ fluxes by eddy covariance and analyse the errors of the method. Most studies conducted over forest show moderate to large in-canopy reactions : Andreae et al. (2002) report comprehensive flux measurements in the Amazonian forest showing evidence of within-forest cycling of the nitrogen oxides emitted from the soil. Horii et al. (2004) report $NO_x$ and $O_3$ fluxes over a temperate deciduous forest showing consistent $NO_x$ deposition. Geddes and Murphy (2014) report such measurements over two mixed hardwood forests in North America, under a very low NO concentrations regime, which shows mainly $NO_x$ deposition with evidence of chemical reactions in the canopy. Min et al. (2014) report such flux measurements over ponderosa pine which shows evidence of within-canopy chemical removal of $NO_x$. Ammann et al. (2012) report total reactive nitrogen fluxes by eddy covariance above grassland which compared well with dynamic chamber NO and $NO_2$ fluxes during periods with no $NH_3$ emissions. Lee et al. (2015) and Marr et al. (2013) report fluxes of NO and $NO_2$ over urban areas which differ in their comparison with national emissions inventories: while Lee et al. (2015) found fluxes 80% higher than national inventories, the second study found similar fluxes but with large disparities at the local scale.

In this study we are addressing the following questions: (1) is the eddy covariance method suitable for quantifying the seasonal dynamics and diurnal cycles of the NO, $NO_2$ and $O_3$ fluxes above a crop rotation? (2) How are organic and mineral fertilisations affecting these fluxes and their dynamics? (3) To what extent are the chemical reactions between NO, $NO_2$ and $O_3$ modifying the fluxes above the ground? And finally, (4) why is $O_3$ deposition increasing following organic fertilisation? Is that a consequence of interactions with NO emissions?

To answer these questions we report measurements of NO, $NO_2$ and $O_3$ fluxes by eddy covariance using a system similar to Lee *et al.* (2015) for one month following slurry spreading over a bare soil at the FR-GRI fluxnet and ICOS site (Loubet et al., 2011). The NO and $O_3$ fluxes were measured over an additional 6 month period to study the seasonality of these fluxes and to measure the fluxes following application of mineral fertiliser.

## 2. Materials and methods

### 2.1 Site description and management

The experiment took place in a 19 ha field located at Grignon (48°51' N, 1°58' E), 40 km west of Paris (France) and lasted more than 7 months from 07/08/2012 to 13/03/2013. The field was surrounded by heavy traffic roads on the east, south and south-west. The field belongs to a large farm (buildings at around 450 m to the south west) with around 210 dairy cows, 500 sheep, and a production of approximately 900 lambs. The terrain has a gentle slope of ~1% and the mean annual temperature and precipitation were 10.9°C and 575 mm between 2005 and 2013. The main wind directions are north-west during clear days and southwest during cloudy and rainy days. The soil type is a *luvisol* or loamy clay (25% clay, 70% silt, 5% sand in the top 15 cm). The soil organic carbon content was ~20 g C $kg^{-1}$, pH (in water) = 7.6, and bulk soil density was 1.3 g $m^{-3}$, in agreement with previous measurements on the same site (Laville et al., 2009 and 2011, Loubet et al., 2011). High pH values are common in soils over calcareous layers and with high fine fraction content (clay and silt) as is the one of the Grignon site. Indeed, alkalinity fosters the nitrification process and this range of pH is optimum for it to occur (e.g. Nieder and Benbi, 2008). A detailed description of the site can be found in Laville et al. (2009; 2011), and Loubet et al. (2011).

The field was cultivated with winter wheat (a mix of Atlass and Premio species), which was harvested on 3/08/2012 (16.7 Mg $ha^{-1}$ of dry matter). Cattle slurry was applied on the field with a trailing hose from the 18 to the 19/08/2012, at a rate of 42 kg N $ha^{-1}$ of which 78% was ammonium ($NH_4^+$). The slurry had a very low dry matter content of 3.2% and a C/N ratio of 15.7. The total C applied was 666 kg C $ha^{-1}$. A gentle tillage was performed on the 29/08/2012 to incorporate the crop and slurry residues and prepare the soil for oilseed rape sowing (variety Adriana) at a density of 35 plants per square meter. The crop developed slowly during the winter with a dry matter above ground (leaf area index) of 37 g $m^{-2}$ (0.65 $m^2$ $m^{-2}$) on the 25/10/2012 and 104 g $m^{-2}$ (0.7 $m^2$ $m^{-2}$) on the 18/02/2013. The canopy height stayed below 10 cm during the whole winter. Ammonium nitrate pellets were applied on the oilseed rape field on the 20/02/2013 at a rate of 54 kg N $ha^{-1}$. Two selective herbicides were applied on the 2 (Springbok: 200 g $L^{-1}$ of Metazachlore and 200 g $L^{-1}$ of DMTA-P at 3 L $ha^{-1}$) and 31/10/2012 (Devin / Cycloxydime: 100 g $L^{-1}$ at 1 L $ha^{-1}$) which only destroyed the weeds. In December 2012 slug repellent pellets were applied.

### 2.2 Micrometeorological and ancillary measurements

Meteorological measurements included wind speed, air and soil temperatures and humidity as well as rainfall, global, net and photosynthetic active radiation. The meteorological measurements were performed on a mast (3.17 m high) near the centre of the field and close to the flux measurement site (Fig. 1). Soil was sampled approximately once a month for water content, total nitrogen and mineral nitrogen analysis. Measurements are described in (Loubet et al., 2011) and will not be detailed here.

A simplified sketch of the EC measurement system is shown in Fig. 1. Three-dimensional wind and temperature fluctuations were measured near the centre of the field at 3.17 m above ground by a sonic anemometer (Gill R3 3-axis anemometer, Gill Instruments Limited, UK). A fast response open-path $CO_2/H_2O$ infrared gas analyser (IRGA LI-7500A, LI-COR, USA) installed at a lateral distance of around 0.2 m to the sonic path measured $CO_2$ and $H_2O$ fluctuations. $O_3$ mixing ratios were measured by a high-frequency, dry chemiluminescence $O_3$ detector (NOAA, USA) and its Teflon PFA inlet tube (length = 2.8 m, internal

diameter = 0.32 mm) was positioned in-between the sonic path and the IRGA at a lower height. The high-
frequency signals were recorded at 20 Hz by a Labview program developed in the laboratory. In accordance with
(Lee et al., 2015), high-frequency (10 Hz) time series of NO and $NO_2$ were determined by two fast-response and
closed-path chemiluminescence NO analysers (CLD 780TR, EcoPhysics, Switzerland), one being coupled to a
photolytic converter (blue light converter, BLC, Droplet Measurement Technologies Inc, USA) for the detection
of $NO_2$ (see Fig. 1). The horizontal separation of the trace gas inlets from the sonic path was 20 cm. Air was
sampled through two heated and shaded PFA tubes with a length of 20 m and an inner diameter of 9.55 mm. The
first CLD was used for measuring NO and the second one connected to the BLC was used for detecting $NO_2$.
Conversion efficiencies for $NO_2$ to NO of around 30% were achieved. The high-frequency signals of NO, $NO_2$
and $O_3$ were calibrated with mixing ratios measured at 30 min time resolution by slow-response analysers
(ThermoScientific, Waltham, USA) (Fig. 1). These instruments were calibrated every 3 to 6 weeks using the gas-
phase titration (GPT) method and a 17 ppm NO standard (Air Liquide, FR). The fetch of the field site extended
at least to 150 m in all directions and a footprint analysis showed that 90% of the time the entire field was in the
footprint during neutral and moderately stable or unstable conditions (Loubet et al., 2011). NO and $O_3$ fast
sensors were functioning during the whole campaign (07/08/2012 to 13/03/2013) together with NO, $NO_2$ and $O_3$
slow-response analysers and the meteorological station. High frequency $NO_2$ measurement was performed from
14/08/2012 to 30/09/2012. In this study we focus on two periods: (1) from 14/08/12 to 29/08/12 during which all
fluxes were measured and NO fluxes were the highest, in order to investigate the interactions between the fluxes
and mixing ratios of the $NO$-$NO_2$-$O_3$ triad, and (2) over the whole period for NO fluxes analysis. **[INSERT**
**FIGURE 1]**
**2.3 Eddy covariance fluxes computations**
The turbulent fluxes were computed as the covariance between the fluctuations of the scalar of
interest and the vertical component of the wind. As the EC method and its theoretical background are described
in the literature - e.g. (Foken, 2008) - details will not be provided here.
For closed-path sensors (NO, $NO_2$ and $O_3$), the lag time between $w'$ and the dry mole fraction $\chi$, had to be
determined. This was done by searching for the maximum of the covariance function
$\overline{(w'(t)\chi'(t-lag))}$.
The lag for NO was 3.1 s [2.4-3.65 s] (Q50 [Q25-Q75]), for $NO_2$ it was 4.0 s [3.65-4.55 s], and for $O_3$ it was
2.9 s [2.5-3.25 s]. The lag was filtered for outliers (points outside of median lag ± standard deviation were
considered as outliers) and the covariance was computed as the value of the covariance function at the filtered
lag.
As fast-response sensors for NO, $NO_2$ and $O_3$ were not absolute, the fluxes were computed following
the ratio method for $O_3$ described by (Muller et al., 2010a), and in accordance with (Lee et al., 2015) for NO and
$NO_2$:
$$F_{O3} = \frac{\overline{\chi_{O3}}}{V_{dry}} \frac{\overline{w'O_3'}}{\overline{O_3}} \qquad (1)$$
$$F_{NO} = \frac{\overline{w'NO'}}{s_{NO}V_{dry}} \qquad (2)$$
$$F_{NO} = \frac{1}{\alpha V_{dry}} \left( \frac{\overline{w'NO_x'}}{S_{NO2}} - \frac{\overline{w'NO'}}{S_{NO}} \right)$$
(3)

where $O_3$(in mV), NO and $NO_x$ (in counts s$^{-1}$) are the uncalibrated fast signals, $\chi_{O3}$ is the 30 min average of the slow-sensor reference $O_3$ mixing ratio (in ppb), while $S_{NO}$ and $S_{NO2}$ are the sensitivity of the analysers (in counts s$^{-1}$ ppb$^{-1}$). $\alpha$ is the blue light converter conversion efficiency, and $V_{dry}$ is the molar volume of dry air (in m$^3$ mol$^{-1}$). All fluxes (momentum, heat, $CO_2$, $H_2O$, NO, $NO_2$, $O_3$) were computed by the EddyPro software-version 5 (www.licor.com/eddypro) and final flux data were averaged for 30 min intervals. Evaluation methodologies from the CarboEurope project were applied - see (Aubinet et al., 2000 ; Loubet et al., 2011).

**2.4 Spectral corrections and flux uncertainties**

Spectral attenuation of the flux is due to differential transport time of the compound in the tube and interaction with tube walls and filter surfaces (Massman and Ibrom, 2008). We tend to minimize this effect by ensuring a large flow rate in the tubes with a Reynolds number well above the critical threshold for turbulence - see (Lenschow and Raupach, 1991) - as well as heating the tubes to around 5°C above ambient temperature. The residence time of the air samples inside the tubing was around 1 s, ensuring low chemical conversions and the Reynolds number was 11700, hence largely in the turbulent range (Re>4000). However, water vapour interaction is still expected, and sensor separation also generates high frequency losses.

The NO, $NO_2$ and $O_3$ random instrument noises were estimated as the 1-$\sigma$ random uncertainty of the signals as in Lenschow (2000), Langford et al. (2015) and Mauder et al. (2008). This is assumed to be "white noise" and hence uncorrelated to itself apart at lag = 0. It is therefore estimated as the difference between the autocorrelation at lag = 0 s and at lag = ± 0.05 s. The flux random uncertainty was itself evaluated as the covariance detection limit. It was determined as the root mean square error of the covariance function over 60 second periods at lag times well away from the position of the time lag. In practice, these were taken at lags larger than 120 s as absolute values as proposed by Langford et al. (2015).

**2.5 Chemical reactions, time scales and flux divergence**

Chemical reactions between NO, $NO_2$ and $O_3$ are important to consider when interpreting the measured fluxes as they can affect the fluxes above the ground. A common way to determine whether these reactions may indeed affect the flux is through comparison of chemical and transport time scales. Details of the reactions rates, time scales and flux divergence calculations are given in the supplementary material sections S1-S3.

**3. Results and discussion**

**3.1 Quality check and uncertainties in NO, $NO_2$ and $O_3$ flux measurements**

NOx and $O_3$ half-hourly fluxes were filtered by the quality check test included in EddyPro (www.licor.com/eddypro), according to the 0-1-2 labelling proposed by Mauder and Foken (2006), that includes tests for stationarity and for well-developed turbulence. As recommended in the framework of the CarboEurope project, we discarded fluxes with a quality check index value of 2. This led to keeping 74%, 84% and 76% of the records for NO, $O_3$ and $NO_2$ respectively. The total records of NO and $O_3$ half-hourly fluxes were 11329 (from 07/08/2012 to 13/03/2013), while for $NO_2$ they were 2257 (during the period 14/08/2012 to 30/09/2012).

The largest systematic uncertainties were the high frequency losses, which were estimated with the in-

situ ogive method (Ammann et al., 2006), and amounted to 10% for $O_3$, 20% for NO and 30% for $NO_2$ on

average over the August-September period (when all fluxes were measured, see Fig. 2). As a bias, they can be

corrected for, and this was performed in the following sections of this manuscript. **[INSERT FIGURE 2]**

The second largest uncertainties were the random uncertainties which were lower than 20% in most

cases for $O_3$, NO (and similar to $H_2O$) and around 30% to 40% for $NO_2$ (Fig. 3). For NO and $NO_2$ the random

uncertainties peaked during the morning traffic hour around 6:00-8:00 UTC, which is explained by the non-

stationarity generated by the local traffic on the mixing ratios. Hence overall the eddy covariance method proved

to be usable for measuring NO fluxes over part of the season with an overall uncertainty similar to $H_2O$. A

higher random uncertainty was found for $NO_2$ fluxes which were smaller than NO fluxes and with a relatively

low conversion ratio from $NO_2$ to NO (30%). **[INSERT FIGURE 3]**

**3.2 Meteorological conditions**

Daily averages of the air temperature decreased during the measurement period, starting from about

20°C in summer and reaching minima around -5°C from December to March. Daily averages of global radiation

decreased from 250 W $m^{-2}$ in August to around 0 W $m^{-2}$ in December, back to around 150 W $m^{-2}$ in the end of

March (Fig. 4). The daily average of the relative humidity was around 65% in August and September, and it

increased to about 85% for the rest of the period. The wettest period was between October and November, and

cumulative rain was 319 mm over the 7 month period, which is quite high. The prevailing wind direction was

south-west while the most intense winds were observed from north and south (Fig. S2). Figure S2 also shows

that wind regimes were quite different in summer and winter: prevailing wind directions during August and

February were from south-west and north-east, respectively. Soil water content (SWC) ranged between 20% and

40% (volume) (Fig. 4), with a long period between October and January with values around 28%, and increased

further in January to 35%, with sharp decrease during some periods. **[INSERT FIGURE 4]**

**3.3 Seasonal dynamics and diurnal cycles of the NO, $NO_2$ and $O_3$ fluxes above a crop rotation**

**3.3.1 Seasonal dynamics of NO-$NO_2$-$O_3$ mixing ratios**

Average daily NO, $NO_2$ and $O_3$ mixing ratios were 3.6, 6.9 and 24.8 ppb, respectively. The NO and

$NO_2$ mixing ratios were higher when winds blew from the east (from the direction of Paris), while $O_3$ showed the

opposite behaviour, which can be explained by depletion of $O_3$ by NO sources from the surrounding traffic (as

shown in Fig. S2) and by reactions (S1-S2). Daily $NO_2$ / $NO_x$ ratios were on average 66%, which is typical for

traffic and urban pollution (Carslaw, 2005; Minoura and Ito, 2010), and ranged from 4% to 93% during the

entire period. The $NO_2$ mixing ratio was significantly higher (Student t-test p-value lower than 8 $10^{-11}$) than the

NO mixing ratios in August and early September, end of January and mid-February, and end of March. During

sporadic episodes, NO peaks were of the same order or even higher than $NO_2$ peaks (Fig. 4).

**3.3.2 Seasonal dynamics of NO $NO_2$ and $O_3$ fluxes**

The daily averaged NO fluxes were very small, except during a period of strong emission following

organic fertilisation over two days in August (18-19/08/2012), with maximum daily average fluxes of around

1.5 nmol $m^{-2}$ $s^{-1}$ (Fig. 4). Other emissions episodes, including mineral fertilisation in February (20/02/2013),

were characterized by mean daily fluxes below 0.5 nmol m$^{-2}$ s$^{-1}$. The NO fluxes were slightly negative for some
events (Q25, Q50 and Q75 equal -0.013, 0.031 and 0.11 nmol m$^{-2}$ s$^{-1}$, Fig. S3). The O$_3$ fluxes ranged between -
13.8 and 0 nmol m$^{-2}$ s$^{-1}$, and averaged to -3.12 nmol m$^{-2}$ s$^{-1}$. The largest O$_3$ deposition fluxes were observed
following organic fertilisation in August, and were correlated with the highest NO emissions. This period also
corresponded to large daily O$_3$ mixing ratios (Fig. 4). The NO$_2$ fluxes were only measured during the first one
and a half months (16/08/2012 to 30/09/2012) and were mostly negative (indicating deposition), except during
the first week following organic fertilisation (Q25, Q50 and Q75 equal -0.11, -0.07 and 0.08 nmol m$^{-2}$ s$^{-1}$) (Fig.
S3). O$_3$ fluxes were in the same range of magnitude, typically between -20 and 0 nmol m$^{-2}$ s$^{-1}$, as those reported
by previous studies at the same site (Stella et al., 2013b; Stella et al., 2011b; Tuzet et al., 2011) and in the
literature over various canopies such as grassland (Stella et al., 2013a), barley (Gerosa et al., 2004), potato field
(Coyle et al., 2009), or forests (Fares et al., 2010; Gerosa et al., 2005), although O$_3$ flux magnitude is sharply
dependent on local O$_3$ mixing ratio. We found similar magnitudes of ozone fluxes in August and September as
those reported by Stella et al. (2013) over a meadow during the summer. We also found similar nocturnal O$_3$
deposition velocity as found by Stella et al. (2011) over bare soil during summer, but with a higher daily
maximum (0.8 cm s-1 instead of 0.5-0.6 cm s-1). Seasonal and daily dynamics of O$_3$ deposition velocity are
shown in Fig. 5. We further find a similar midday magnitude as Stella et al. (2011) found in April with wetter
soils. Night-time ozone deposition velocity did not go lower than around 0.2 cm s$^{-1}$ in our study, as also found by
Zhu et al. (2015) over a growing wheat in China, Stella et al. (2011) over bare soil in summer, and Lamaud et al.
(2009) over maize. These authors as well as Huang et al. (2016) clearly show that this is due to non-stomatal
deposition being primarily driven by $u_*$ which does not reach zero at night during these periods. We can hence
conclude that we found consistent ozone deposition in August and September compared to other studies at that
site or in other geographical areas. When compared to previous years at the same site the deposition velocity
measured during the winter in this study was clearly smaller. We interpret this as being primarily due to $u_*$ being
smaller that winter compared to other winters, as well as due to a bad development of the winter crop due to soil
drought in September (SWC =20% in the 15 cm horizon).
**[INSERT FIGURE 5]**

**3.3.3 Comparison of ozone fluxes to the Stella et al. (2011)  parameterisation over soil**

In order to compare to previous studies of ozone deposition onto bare soil on the same site, we have
calculated the soil surface resistance as in Stella et al. (2011) and deduced the ozone deposition velocity as
$V_{dO3} = (R_{soilO3} + R_{bO3} + R_a(z_{ref}))^{-1}$. In this way, we can compare the two studies while excluding any confounding
factors (roughness and turbulent exchange intensity). We can see in Fig. 6a that the measured ozone deposition
velocity during August follows the parameterisation of Stella et al. (2011) most of the time except for some days
including 18 and 19 August which corresponds to slurry application and 24, 25, 26 August, which follows a
small rainfall. We also see an overestimation of the Stella parameterisation before the 18 August, which we
interpret as being due to the straw and wheat residues being present on the ground before slurry incorporation.
This comparison hence demonstrates that the ozone deposition was indeed increased slightly following slurry
application and subsequently following rainfall. This may be either due to a physical reason (increased surface
exchange on the soil due to tillage or humidity change due to slurry) or a chemical reason (surface reactivity
changes due to added organic matter or VOC emissions from the slurry). Fig. 6b further shows that the main
differences are observed for wet soils and relatively low temperatures (this is after rainfall) and to a lesser extent
for dryer and hotter situation (following slurry spreading).
**[INSERT FIGURE 6a and 6b]**

### 3.3.4 Diurnal cycles of mixing ratios and fluxes over periods of interest

$O_3$ mixing ratios exhibited a typical diurnal cycle that was governed by photochemistry and convective
mixing within the boundary layer and from the free troposphere during daytime. It started to increase with
sunlight around 7 a.m., and declined in the evening starting from 6 p.m. due to lack of photochemical formation
in the absence of sunlight, as well as deposition and destruction with NO in this high $NO_x$ emission area. In
general, NO mixing ratios featured a marked peak in the early morning and remained high until around 13:00
UTC (Fig. 7b). During the early afternoon, the $O_3$ increase was correlated with the NO decrease. $NO_2$ mixing
ratios showed a bi-modal diurnal cycle with its maxima in correspondence with morning and evening traffic
peaks, i.e. around 6 a.m. and 7 p.m..
The NO fluxes also showed a diurnal cycle similar to the one of soil temperature with an emission peak
around 12 a.m. (Fig. 7a and b). This suggests that NO emissions are related to nitrification, for which the
emission rate is an exponential function of soil temperature (Henault et al., 2005). This was already shown for
the Grignon soil by Laville et al. (2011). The fact that NO fluxes decrease earlier than soil temperature is most
likely due to titration of NO by $O_3$ in the late morning and early afternoon, causing the NO emissions at the
reference height to be reduced with respect to ground emissions. After correction for chemical reactions the NO
emissions diurnal cycle is indeed at a peak later in the day, more in phase with ground temperature (see Fig. 11).
This is also indicated by the positive $NO_2$ flux observed during the same time of the day. The $O_3$ flux was mainly
negative (deposition) and follows the diurnal dynamics of measured mixing ratios. **In terms of deposition**
**velocity, the ozone deposition velocity followed a clear diurnal cycle with a maximum during the day and a**
**minimum at night. The measured $NO_2$ deposition velocity showed slightly negative values, but slightly**
**positive ones when corrected for reactions with NO and $O_3$.**
**[INSERT FIGURE 7a and 7b]**

### 3.4 Influence of organic and mineral fertilisations on NO emissions

The NO flux averaged over the whole period was 0.09 nmol m$^{-2}$ s$^{-1}$ (mean), which is in the range of
previous findings for the same site. Laville (2011) and Loubet et al. (2011) reported yearly averaged NO fluxes
varying between 0.07 and 0.15 nmol m$^{-2}$ s$^{-1}$ for 2007-2009. The NO flux distribution was shifted towards
positive values after the organic fertilisation in August (Fig. S3), with the mean NO flux during the two weeks
following the fertilisation (0.49 nmol m$^{-2}$ s$^{-1}$) being six times larger than the one for the whole period. For the
same period, the ozone flux distribution was shifted towards more negative values. Figure S3 also shows that
flux distributions after mineral fertilization do not differ much from the ones relative to the whole period. During
the two weeks following the February mineral fertilisation the NO flux increased less and was only 1.7 larger
than over the whole period (0.14 nmol m$^{-2}$ s$^{-1}$). These numbers are also in line with those reported following
fertilisation on the same soil in the 2007-2009 period by Laville (2011) and Loubet et al. (2011), which also
showed some periods with slightly negative NO fluxes. Stella et al. (2012) measured a larger peak of NO

emissions following slurry spreading, but only lasting two to three days, which was probably due to a dryer soil in our study compared to Stella et al. (2012).

Following the slurry application, the NO emissions amounted to 0.1 kg N ha$^{-1}$, which represents 0.24% of the applied nitrogen (42 kg N). Following the mineral fertilisation, the NO emissions amounted to 0.02 kg N ha$^{-1}$, which represents 0.037% of the applied nitrogen (54 kg N). Over the whole period from August 2012 to March 2013, we evaluate a loss of 0.26 kg N ha$^{-1}$. With a total N input of 96 kg N ha$^{-1}$, this gives an estimate of the NO emission factor of 0.27%, which is similar to values reported earlier for the same site (Laville et al., 2011) but one order of magnitude larger than the EMEP/IPCC default value of 0.04. Nevertheless, this is an average value calculated with the Tier 1 approach, which does not take into account correction factors depending on soil pH or fertilizer type. This more detailed approach, the Tier 2, has not been developed for NO (EEA, 2016).

The reasons for lower emissions following winter mineral fertilisation than following summer manure application are manifold. Even if the amount of applied nitrogen was similar for the two cases (42 and 54 kg N ha$^{-1}$), meteorological and soil conditions were much more favourable for nitrification in summer than in winter (Davidson, 1992; Williams and Fehsenfeld, 1991). Indeed, NO emissions from agricultural soils are primarily the by-products of nitrification, and this hypothesis was tested for the Grignon site by Laville et al., (2011). Nitrification is inhibited by low soil temperature and high water content that causes anoxia. Soil temperature was much lower in February than in August (2.5 compared to 20 °C on average). February was particularly humid, with a total precipitation of 10 mm, while in August no significant rain event occurred after the first week. In this period indeed, the soil was only humidified by the organic manure supply (on a 4.8 mm thick layer ) that was applied on a dry soil. The soil water content at 5 cm depth in September 2012 was around 21% in volume, while in February it was 33% in volume. These two factors led to more favourable conditions for nitrification in August than in February.

**3.5 Influence of surrounding roads on the measured fluxes and concentrations of the NO-NO$_2$-O$_3$ triad**

Using the FIDES flux and concentration footprint model (Loubet et al., 2010) we evaluated the footprint of nearby roads. Overall the flux footprint from the nearby roads was smaller than 1% (which means that only 1% of the road emissions contributes to the flux at the mast) most of the time, but the concentration footprint reaches up to 10% during some episodes, with separate roads contributing differently depending on the period (Fig. S1). Assuming a conservative emission of 250 mg km$^{-1}$ vehicle$^{-1}$ and an average vehicle count of 10000 vehicles per day (2010 counts, "Statistiques du département des Yvelines pour 2010" ranges between 5000 and 15000), we evaluate that the roads contribute from 4% to 40% to the measured fluxes. However, since vehicles emissions of NO$_x$ have a sporadic nature, 10000 vehicles per day means a maximum of ~1 vehicle every 2 seconds (if we consider, conservatively, that most of the traffic is condensed during 9 hours only). These vehicles are also moving at about 90 km h$^{-1}$ (25 m s$^{-1}$), hence leading to a moving point source of NO$_x$. We therefore expect that the signal of this moving and sporadic source is not captured by the eddy-covariance method, and would be filtered out by despiking and flux calculation procedures (Foken, 2008 ; Mahrt, 2010).

## 3.6 Chemical interactions: the NO-NO$_2$-O$_3$ triad and effect on the fluxes

In order to investigate the interactions between the fluxes and mixing ratios of the NO-NO$_2$-O$_3$ triad, we focus on the period from 14/08/12 to 29/08/12, during which all fluxes were measured and NO fluxes were the highest.

The two weeks following the organic manure application (from 18/08 to 19/08) are characterized by hot sunny days, with maximal global radiation above 800 W m$^{-2}$, except for 24/08 when the only rain event occurred (Fig. 8). The period of August 18th to the 23rd was the warmest, with soil surface temperatures above 40°C at noon during most days, while the air temperature decreased from around 35°C to around 20°C during the same period. The soil temperature at 5 cm depth followed the same trend, but with a lower daily maximum and a higher night-time minimum. Due to sensor breakdown the soil water content was not measured during this period. The small latent heat flux (LE) after the 19th of August, (17 W m$^{-2}$ on average between August 19th and the 31st) the large sensible heat flux (60 W m$^{-2}$ on average) and radiation (212 W m$^{-2}$ on average) indicate that the soil humidity of the top soil layer was low. Hence, we assume that the SWC was probably similar to what was measured in September (around 20 % in volume), which is ideal for nitrification to occur (Laville et al., 2011; Oswald et al., 2013 ). **[INSERT FIGURE 8]**

The 18/08 was the first day when NO emissions from the soil occurred. The emissions lasted around two weeks following the organic fertilisation (Fig. 4), during which the NO flux during daytime exceeded 0.5 nmol m$^{-2}$ s$^{-1}$, peaking around 12 a.m. The nocturnal NO flux usually decreased to zero, except for the night of 25/08, characterized by strong winds (Fig. 8). The maximum of the NO emissions was 2.7 nmol m$^{-2}$ s$^{-1}$ observed six days after fertilisation on 21/08.

The NO$_2$ flux daily pattern was different during the two weeks following organic manure application compared to the period before (Fig. 8). It was in general positive during the day and around zero at night during the period from 18/08 to 29/08, except for the night of 25/08 when it was large and negative. Positive NO$_2$ fluxes might be explained by chemical reactions between NO and O$_3$ in the surface layer (De Arellano et al., 1993), which will be discussed in the next section.

The O$_3$ flux was also significantly higher (Student t-test p-value lower than 2 10$^{-16}$) following organic fertilisation (mean -10.7 nmol m$^{-2}$ s$^{-1}$) than during the rest of the experimental campaign (mean -3.1 nmol m$^{-2}$ s$^{-1}$) (Fig. S3). Since the mixing ratio of O$_3$ was quite variable during the campaign (Fig. 4), it is more interesting to look at the deposition velocity which underpins the surface exchange processes (Fig. 7b and 8). The median $V_{dO3}$ during the organic fertilisation event exceeded the median over the rest of the experimental campaign by a factor of two. However, this increase in O$_3$ deposition velocity cannot be explained by reaction with soil-emitted NO alone as the O$_3$ flux is an order of magnitude larger than the NO flux.

Different pathways for the near-surface O$_3$ removal are likely: i) photolysis of O$_3$ by ultraviolet light in the presence of water vapor forming OH radicals, ii) gas phase reactions with reactive VOCs and iii) heterogeneous reactions with the soil or with molecules adsorbed on soil..

The NO mixing ratio was well correlated with the NO flux, with a normal correlation coefficient of 40% for the two weeks following the organic fertilisation (excluding 24-25 August), while it was only 2% for the 7 month period. This suggests that, following fertilisation, the ambient NO levels were mainly due to local emissions. The NO$_2$ mixing ratio was less correlated with the NO$_2$ flux, suggesting that NO$_2$ levels were more related to advection from surrounding road traffic than from local emissions. Indeed, both NO and NO$_2$ are

emitted from road traffic and urban pollution, but the $NO_2$ component quickly becomes prevalent as the plume is advected, especially in presence of high $O_3$ levels, as in our case (Carslaw, 2005; Minoura and Ito, 2010). The minimum night-time mixing ratio is mainly controlled by night-time wind velocity: the higher the night-time velocity, the higher the mixing ratio, due to a better mixing in the atmospheric surface layer. During conditions with lower wind speed, deposition and reaction with local $NO_x$ sources lead to a high depletion of $O_3$ during the night.

**3.7 To what extent are the chemical reactions between NO, $NO_2$ and $O_3$ modifying the fluxes above the ground?**

Measured mixing ratios and fluxes of NO, $NO_2$ and $O_3$ are affected by chemical reactions (S1 to S4) in addition to emissions and deposition processes. Especially, the diurnal fluxes of $NO_2$ observed from the 18 to the 23 of August, were positive (emissions) and of the same order of magnitude as the NO fluxes, while they were negative afterwards. The simultaneous observation of positive NO and $NO_2$ fluxes are typical for the NO-to-$NO_2$ transformation below the flux observation level in the presence of high $O_3$ mixing ratios. This phenomenon is called "apparent $NO_2$ emissions" and was observed in other studies mainly above dense or tall canopies (Ammann et al., 2012; Min et al., 2014; Plake et al., 2015). For the reactions (S1-S2) to occur below the measurement height, the turbulent transport time ($\tau_{trans}$) needs to exceed the chemical reaction time ($\tau_{chem}$) (Arellano and Duynkerke, 1992; De Arellano et al., 1993; Lenschow and Delany, 1987; Plake et al., 2015; Stella et al., 2013a; Stella et al., 2011a; Stella et al., 2012). The Damköhler number Da = $\tau_{trans}$ / $\tau_{chem}$ is often used to determine the conditions favourable for chemical reactions: in cases when Da is higher than unity chemical reactions are faster than the transport (flux divergence), whereas Da values smaller than 0.1 indicate that the influence of chemical reactions was negligible. The aerodynamic resistance $R_a(z)$ (Eq. S8) was overall quite small and ranging from 45 to 128 s m$^{-1}$ (1$^{st}$ and 3$^{rd}$ quantiles), hence leading to a quite short transport time scale (but larger than 100 s most of the time). The boundary layer resistance was around 22 and 43 s m$^{-1}$ (1$^{st}$ and 3$^{rd}$ quantiles) for $O_3$ (Fig. 9). The surface resistance for $O_3$ was estimated as $R_{soil}(O_3) = V_{dO3}^{-1} - R_a - R_b(O_3)$, and dominated the other resistances (100 to 480 s m$^{-1}$). The $O_3$ penetration depth in the soil was estimated as the depth necessary to explain the measured $R_{soil}(O_3)$ if molecular diffusion in the soil pores is the main limitation factor. In practice this corresponded to the dry soil layer used in (Personne et al., 2009). This depth ranged from 2 to 10 mm on average and was smaller at noon than during the night (Fig. 9). Overall, the chemical time $\tau_{chem}$ and the transport time $\tau_{trans}$ were of the same order of magnitude at any time of the day between applications and during mineral fertilisation, and $\tau_{chem}$ was smaller than $\tau_{trans}$ during the organic fertilisation. As a consequence, the Damköhler number was around unity most of the time and larger than unity during the organic fertilisation period, showing that the reaction between $O_3$, NO and $NO_2$ happened during transport from the ground to the EC measurement height at all times at this site. During the fertilisation event, the Damköhler number was especially high at night, when the transport time increased more substantially than the chemical timescale. These results are similar to findings by Stella et al. (2012) for the same site over bare soil. During the periods with vegetation, the increase of the transport time scale above the canopy was less than that of the chemical time scale during nighttime, as the presence of vegetation increases the mixing, and, hence diminishes $R_a(z)$. **[INSERT FIGURE 9]**

The Damköhler number shows that NO reacts with $O_3$ and that photolysis also plays a role. How does
this affect the NO flux measured at the reference height compared to the one at the ground? We quantified this
variation by numerically solving Eq. S13, based on the model of Duyzer et al. (1995). Due to the reaction with
$O_3$, the calculated NO flux at the ground surface was on average 32% larger than that at the measurement height
during the period 17-29/08 (0.93 instead of 0.63 nmol m$^{-2}$ s$^{-1}$). This would represent an increase of 37 g of N
emission following slurry spreading. For $NO_2$, the calculated flux at the ground surface was mostly negative
while it was mainly positive at the reference height during the period 18-22/08. On average the $NO_2$ flux at the
ground was -0.33 nmol m$^{-2}$ s$^{-1}$ over the period 17-29/08 while it was -0.03 nmol m$^{-2}$ s$^{-1}$ at the reference height.
For NO fluxes, the major discrepancy between fluxes at the surface and the measurement height occurs during
periods with relatively large and stable values of the Damköhler number (Fig. 10), as this is the case when
chemical reactions consume NO before it reaches the measurement height.

**[INSERT FIGURE 10]**


The derivation of surface fluxes with the Duyzer model also leads to a diurnal cycle of the NO flux that is closer
to the one observed for ground temperature, corroborating the hypothesis that ground emissions are mostly due
to nitrification for our site (Fig. 11).

**[INSERT FIGURE 11]**


Since the $O_3$ deposition flux was much larger than the NO flux, the reaction with NO changed the

absolute value by only 3% when comparing the flux at the measurement height to the ground surface. Indeed, as
only reactions (S1) and (S2) are considered in eqs. (S12) and (S13), which we used to numerically evaluate
surface fluxes, we obtain: $\Delta[FNO] = \Delta[FO_3] = -\Delta[FNO_2] = 0.3$ nmol m$^{-2}$ s$^{-1}$ where $\Delta$ stands for the
difference between surface and measurement height.
**3.8 Why is $O_3$ deposition increasing following organic fertilisation?**
We observed that following organic fertilisation (performed by injection and hence soil tillage), $O_3$ deposition
increased by a factor of two (as shown by the deposition velocity, Figs. 9 and 10). Several hypotheses may
explain this increase: (1) the increased surface exchange due to soil tillage, (2) the reaction with NO emitted by
the ground, and (3) the reaction with VOCs emitted by the ground:
A first hypothesis (1) would be that the increase in deposition velocity following the organic fertilization could
be due to a change in physical characteristics of the soil surface. Indeed, the application of cattle slurry with a
trailing hose modifies the soil structure at the surface which consequently increases the available surface for $O_3$
deposition, and therefore the deposition velocity. This hypothesis is consistent with the comparison of measured
deposition velocities and modelled deposition velocities using the Stella et al. (2011a) $R_{soil}$ parameterization (see
sect. 3.3.3 and Figure 6a). Indeed, while there is a good agreement between measured and modelled $V_d$ after the
26$^{th}$ of August (i.e., after the rainfall event), modelled $V_d$ systematically underestimates measured $V_d$ between
slurry application and the rainfall event. Since the parameterization of $R_{soil}$ was obtained for the Grignon site
over different periods, that means $R_{soil}$ accounts for the mean soil structure of the Grignon site. Therefore, it can
be hypothesized that (i) $R_{soil}$ is underestimated from slurry application to the rainfall event due to the change of
soil surface structure, and (ii) after the rainfall event, the soil surface recovers its mean structure corresponding
to the $R_{soil}$ parameterization.
A second hypothesis (2) would be that $O_3$ would react with NO emitted by the soil. Although the reactions with
NO during transport are shown to be small compared to the NO flux (Figure 10), reactions in the soil surface
layer may be more significant due to large NO concentrations in the soil, despite the fact that this layer is very
small. A graph of the difference between the measured and the modelled ozone flux following fertilisation (Fig.
S4) seems to show that the additional $O_3$ deposition is correlated with the NO flux. This would mean that the
$NO_2$ produced in the soil by reaction with NO would be adsorbed on the soil surface either in the mineral phase
or dissolved in the water phase as $NO_2$. To evaluate this assumption further, we evaluated the Damköhler
number in the soil surface layer by assuming that the layer depth is equal to the $O_3$ penetration depth $\delta_{O3soil}$ (Fig.
9). In this layer the transport time is equal to soil resistance for $O_3$ times the penetration depth $R_{soilO3} \times \delta_{O3soil}$. We
can evaluate the NO mixing ratio that would explain the additional $O_3$ destruction at the surface, by searching for
the value of $[NO]_{soil}$ that satisfies $\tau_{trans}(Soil, O_3) = \tau_{chem}(Soil, O_3)$. By doing so, we found that $[NO]_{soil}$ would
need to reach 5 to 40 ppm to explain the increase in $O_3$ deposition following organic fertilisation. Gut *et al.*
(1998) and Gut *et al.* (1999) measured NO mixing ratios at a 2 cm depth in the soil under wheat with the
membrane tube technique and report mixing ratios around 100 ppb and always below 400 ppb following
fertilisation, which is one to two orders of magnitude below the mixing ratio which would be needed to explain
the observed $O_3$ flux. Moreover, the rate of NO production in the soil surface layer would have to be equal to the
$O_3$ flux to the ground (around 20 nmol m$^{-2}$ s$^{-1}$) which is an order of magnitude larger than what Gut *et al.* (1998)
or Laville *et al.* (2009) report as maximum NO flux. However we should stress that both Gut *et al.* and Laville *et*
*al.* report NO fluxes that were measured in the presence of ozone and hence would have been depleted by
reaction with it in a similar way as here.
A third hypothesis would be that $O_3$ would react with VOCs emitted by the ground. Reactive VOCs such as
sesquiterpenes and monoterpenes were previously found to be emitted from soils (Horvath et al., 2012; Penuelas
et al., 2014), and some of these sesquiterpene species react with $O_3$ in the order of a few seconds. The reactions
of $O_3$ with larger terpenes are important sources of OH, as well as the ozonolysis of simpler unsaturated
compounds. (Donahue et al., 2005). Currently, there is little or no data available on the emission of VOCs from
slurry application. However, a recent study mainly focusing on quantification of odor emissions from soil
application of manure slurry, showed the formation of a certain number of VOCs, included organic sulfur
compounds, carboxylic acids, alcohols, carbonyl compounds (ketones and aldehydes), aromatic compounds
(phenols and indoles) and nitrogen compounds (Feilberg et al., 2015). Based on their analyses, the compound
most responsible for the overall odor impact from the VOC emissions was 4-methylphenol. These authors also
showed the emission of trimethylamine, a compound that can react quickly with $O_3$, leading to formation of
secondary organic aerosols (Murphy et al., 2007). Furthermore, these authors suggest that a large part of these
VOCs are formed through ozonation reactions (i.e. byproducts of ozonation: methanol, acetone, and
acetaldehyde). Indeed, the slurry would be transported downwards through the soil, where efficient
heterogeneous reactions can take place at particle interfaces. It has been shown that the heterogeneous reaction
probabilities may be much greater than anticipated. For example, measurements on oxide surfaces with a
chemical structure commonly found in VOCs (i.e. alkenes, terpenes, carbonyls) showed that the $O_3$ reaction
probability of a surface-attached alkene can be up to five orders of magnitude greater than for the same reaction
in the gas-phase (Stokes et al., 2008). In the same way, Fick et al. (2005) observed that ozonolysis reaction rates
of some terpenes were much higher than predicted, possibly as a result of reactions on the surfaces used in their
experiments. These results suggest that terpenes can remain on the surfaces, enhancing the $O_3$ reactivity.
Similarly, some other authors observed that surface reaction probabilities with $O_3$ were 10 to 120 times greater
than their corresponding gas-phase values (Dubowski et al., 2004; Springs et al., 2011). It is also known that
soils can act as a sink of VOCs, by their adsorption to soil mineral particle surfaces and humic substances
(Penuelas et al., 2014). Hence, it is likely that surface chemistry including photo-enhanced $O_3$ uptake on organic
matter (Jammoul et al., 2008; Reeser et al., 2009) may explain the increase in $O_3$ deposition, a process not yet
described in the literature. It may also be likely that $O_3$ is destroyed by very reactive VOCs in the gas phase as
hypothesized by Wolfe et al. (2011). These gas-phase reactions, however, require that the chemical reaction time
to be shorter than the turbulence transport time (Plake et al., 2015; Stella et al., 2012).

However, our study does not allow us to conclude definitively which of the three hypotheses is the most likely.
**4. Conclusions**
Eddy covariance flux measurements of the $NO$-$NO_2$-$O_3$ triad during a 7 months period allowed
evaluating several mechanisms controlling the exchange of these reactive trace gases with an agricultural soil.
Eddy covariance technique proved to be suitable at capturing seasonal and diurnal dynamics of the fluxes, and
allowed to interpret flux behaviour according to meteorological variables, fertilisation practices and chemical
reactions. Nevertheless, random uncertainty was particularly important (>20%) during morning traffic peaks due
to non-stationarity of $NO_x$ and $O_3$ mixing ratios. As concerns $NO_2$, uncertainty was even higher (up to 40%) due
to the indirect measurement method. We thus recommend caution in the use of the method in non-stationary
conditions, and combined measurements of horizontal gradients of mixing ratios to quantify the effect of
advection. Also, additional measurements of surface mixing ratios would be useful to check the reconstruction of
surface fluxes that we performed by using the logarithmic-profile model of Duyzer. Finally, high $NO_2$ to $NO$
conversion efficiency should be assured to reduce uncertainty of $NO_2$ fluxes.
In particular, the magnitude and temporal variability of $NO$ emission fluxes following two fertilisation
episodes were analysed, one in summer and the other one in winter. Mean $NO$ emissions during the whole period
were in agreement with previous studies on the same site. Emissions were significantly higher (Student t-test p-
value lower than $2 \times 10^{-16}$, and a factor of seven difference on the mean) during two weeks following organic
fertilisation in August than during the rest of the experimental period. These large emissions are mainly due to
favourable conditions for nitrification: soil water content around 20% and high temperatures. In February,
following mineral fertilisation, the increase of $NO$ emissions was less pronounced, although the same amount of
N was applied. This difference is likely due to less favourable conditions for nitrification in February (low
temperature and higher soil water content), rather than to the different form of fertilizer. On average over the
whole period, we derived a loss of 0.26 kg N ha$^{-1}$ as $NO$ from the field. With a total N input of 96 kg N ha$^{-1}$, this
results in an $NO$ emission factor of 0.27%, which is in the lower range of earlier reported values on this site
(Laville et al., 2011).
Our findings show that $NO$ emissions from agricultural soils are limited (0.27% of the N-$NO$ applied
over the 7 month period, which with a conservative estimation can be extended to a yearly amount). When
hypothetically extended to France with an average nitrogen fertiliser use of 80 kg N ha$^{-1}$ over a fertilised area of
around 26 Mha, this would lead to a total emission of $NO_x$ of around 5.6 kt N-NO. This is negligible compared
to national emissions which are around 240 kt N-NO (CITEPA, 2015). The seasonality and spatial distribution of
these emissions may, however, lead to air quality issues during spring and late summer-autumn which are the
main fertiliser application periods in rural environments. Indeed, most of the emissions we measured occurred
within a few weeks following fertilisation. In terms of ozone, our findings are in accordance with previous ones,
showing that ozone is efficiently deposited throughout the year. This means that crops are participating through
this process in the reduction of the atmospheric oxidising capacity.

The $O_3$ deposition velocity was significantly higher following organic fertilisation than during the rest
of the experiment (Student t-test p-value lower than $2 \cdot 10^{-16}$ and a factor 3 difference on the mean), despite the
fact that vegetation was absent. This increase in $O_3$ deposition could not be explained by the reaction of $O_3$ with
NO in the atmosphere as the NO flux was an order of magnitude smaller than that of $O_3$. The process behind this
ozone deposition increase remains to be discovered. We hypothesised three underlying processes: (1) increase in
soil surface due to soil tillage, (2) reaction with NO in the soil layer, and, (3) reactions of $O_3$ with VOCs emitted
by the slurry. None of these hypotheses can be dismissed and further investigation is required. Experiments in
controlled conditions are desirable to better understand these processes.

The evaluation of the chemical and turbulent transport times showed that reactions between NO, $NO_2$
and $O_3$ below the measurement height occurred during the whole measurement period, leading to a depletion of
NO and a build-up of $NO_2$ from the ground to the measurement height. Following organic manure application,
NO fluxes were reduced by 30% from the surface to measurement height, while the $NO_2$ fluxes switched from
deposition to uptake, being negative at the surface and positive at the measurement height. This phenomenon of
"apparent $NO_2$ emissions" was reported in other studies, especially above forests. Here it also appears to be
important above a bare soil and at moderate measurement heights, during conditions of strong NO emissions and
high ambient $O_3$ mixing ratios.
**Acknowledgements**
This work was funded by the FP7 projects ECLAIRE (grant number 282910) and INGOS (grant agreement
284274), the French ANR project ANAEE, as well as ICOS France. The authors acknowledge the director of the
AgroParsiTech Farm Dominique Tristan for allowing access to the field. We are grateful to the Max Planck
Institute for Chemistry (Mainz, Germany) for the loan of a CLD 780TR analyser for the duration of the
experiment. We also thank Gerardo Fratini for the precious support on issues concerning EddyPro, and Christof
Ammann and Veronika Wolff for fruitful discussion about $NO_x$-$O_3$ chemical reactions. We also thank Polina
Voylokov for her thorough correction of the manuscript and Anaïs Durand for helping on NO emission
inventory evaluations.

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

**Figures**

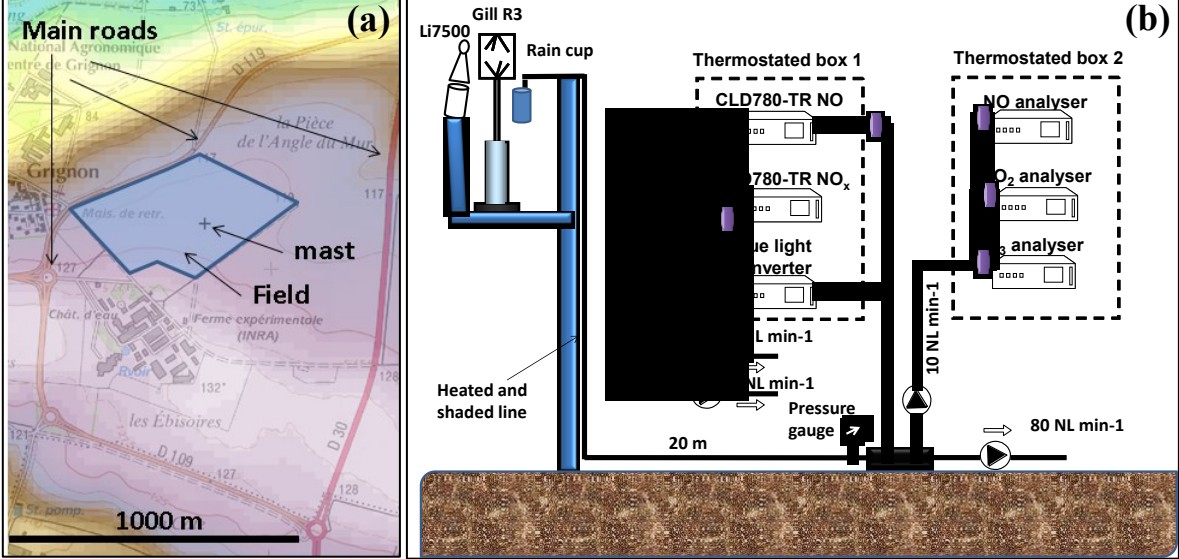


**Figure 1. (a) Simplified map of the FR-GRI field site showing the mast and surrounding roads. The collors**
**correspond to elevation. (b) Simplified sketch of the instrumental setup to measure EC fluxes. Gill R3 is the ultrasonic**
**anemometer, Li7500 is the open path infrared $CO_2$ and $H_2O$ gas analyser, the rain cup is the air sampler for NO and**
**$NO_2$ detection. CLD780-TR NO and NOx are the fast-response NO analysers (Ecophysics) operating in parallel, one**
**connected to a BLC measuring NO + $\alpha NO_2$. The NO, $NO_2$, and $O_3$ slow analysers (ThermoScientific, Waltham, USA)**
**are placed behind a Teflon pump ensuring atmospheric pressure at the inlet. A large pallet pump ensured a flow rate**
**of 80 NL min$^{-1}$ in the heated inlet line. Teflon filters (1μm) were installed at the front of the instrument inlets (purple**
**cylinders).**

Correction factor

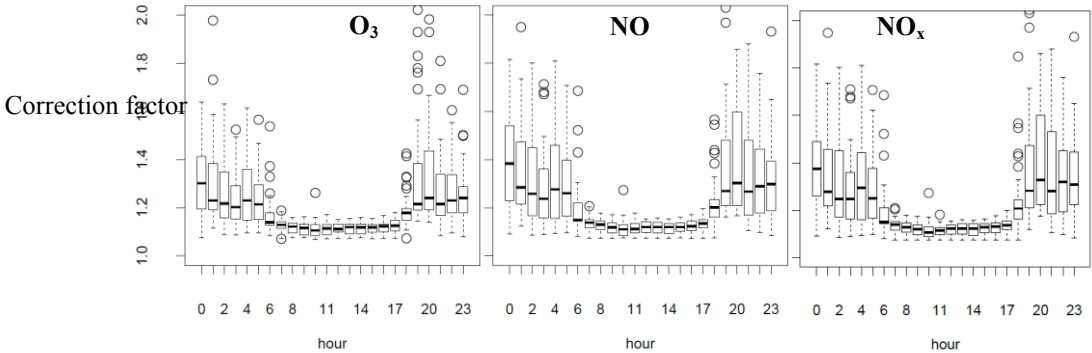


**Figure 2. Hourly averaged high frequency loss correction factors for $O_3$, NO and $NO_x$ over the 15/08/2012 07/09/2012**
**period determined with the in situ ogive method. The corrected flux equals the measured flux multiplied by the**
**correction factor. Black bars are medians, boxes show the interquartile, error-bars show the minimum and maximum**
**of the whisker and empty dots shows the outliers.**

Randomn uncertainty

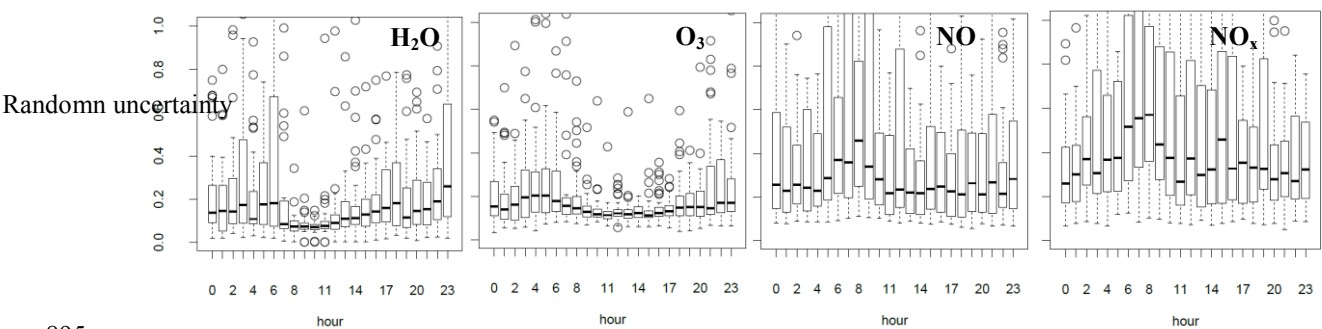

**Figure 3. Daily variations of the ratio of the random uncertainty to the flux for $H_2O$, $O_3$, NO and $NO_x$ during august**
**2012 (15/08 to 09/09). Black bars are medians, boxes show the interquartile, error-bars show the minimum and**
**maximum of the whisker and empty dots shows the outliers.**


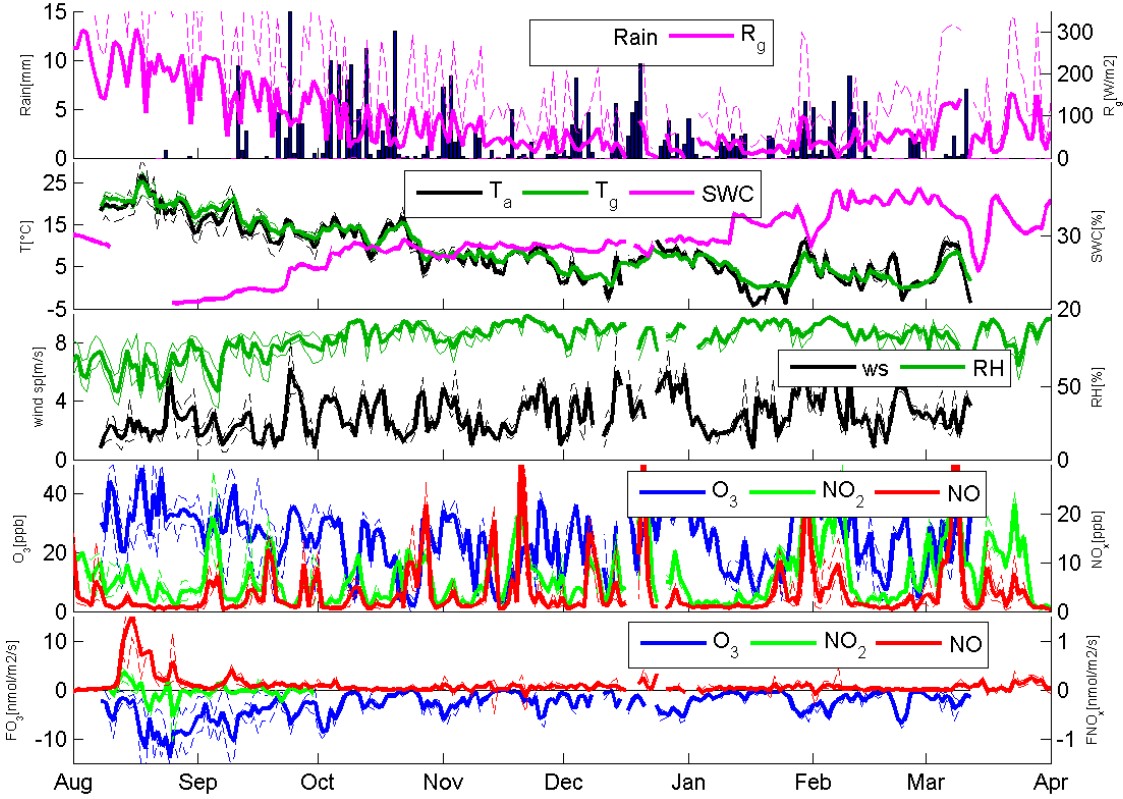


Figure 4. Meteorological and soil conditions (daily averages, sums for rainfall), NO, NO₂ and O₃ mixing ratios and fluxes during the entire measurement period from 07/08/2012 to 13/03/2013 at the Grignon field site. Averages for night-time and daytime are also given as dotted lines. $R_g$ is the global radiation, $T_a$ and $T_g$ the air and ground temperature, SWC the soil water content, ws the wind speed, RH the air relative humidity.


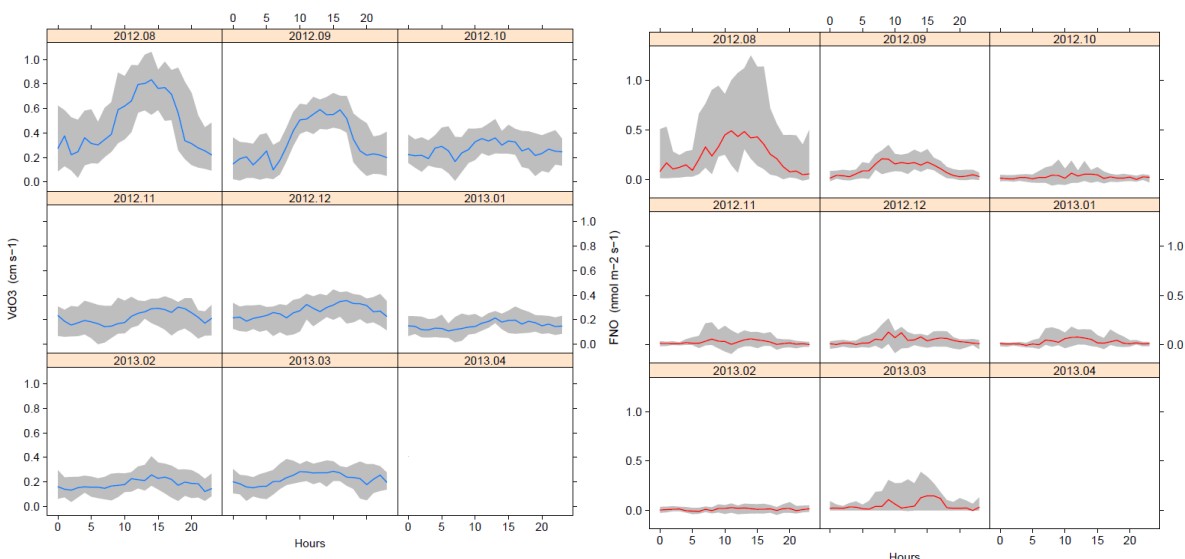


Figure 5. Seasonal changes of ozone deposition velocity VdO3 and NO fluxes. Blue lines show median and grey area inter-quantiles.

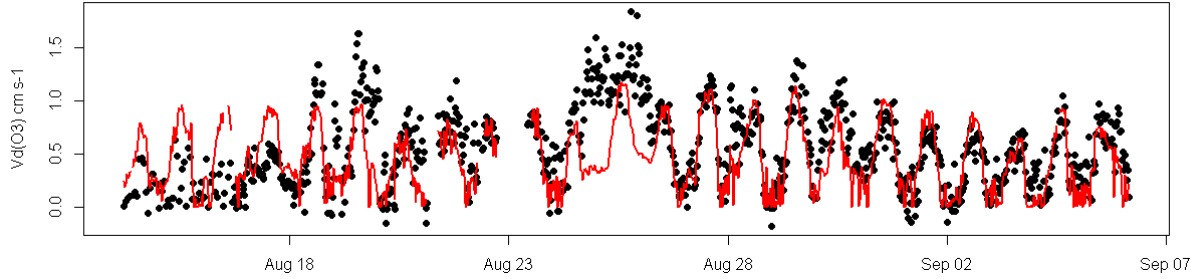


**Figure 6a. Comparison of ozone deposition velocity from this study (black dots), and from the parameterisation of Stella et al. (2011) (red line) based on surface temperature.**

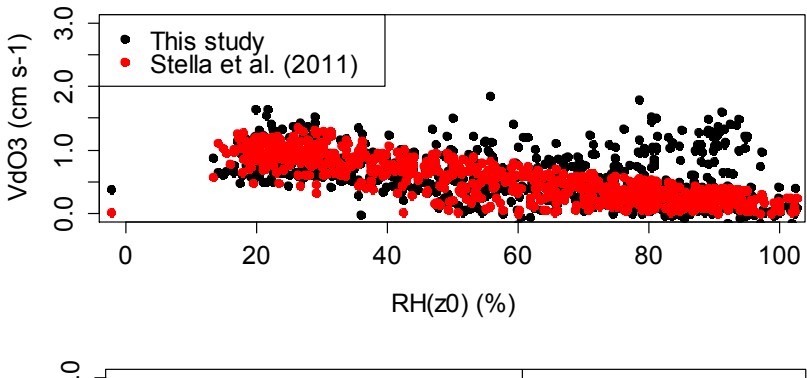


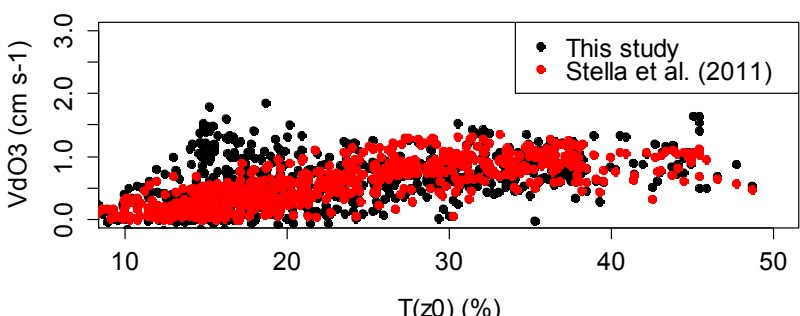


**Figure 6b. Response of ozone deposition velocity to surface humidity $RH(z_0)$ and surface temperature $T(z_0)$. Shown are data from this study and from the parameterisation of Stella et al. (2011). Period from 14 August to 6 September which is before and after slurry spreading and corresponds to Figure S5.**



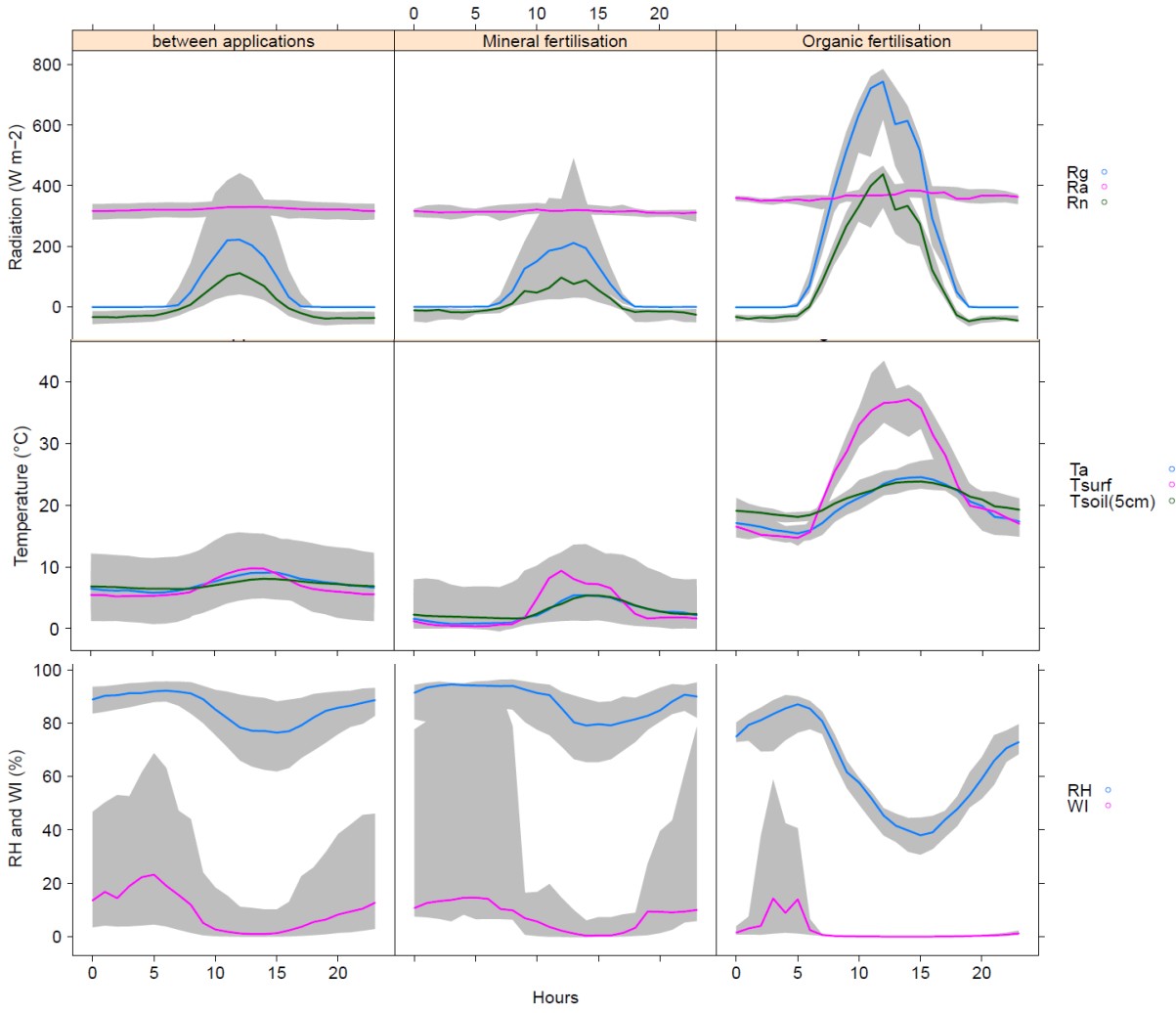

**Figure 7a. Diurnal cycles of global irradiance and net radiation, air and soil temperatures, relative humidity and**
**wetness index averaged over the three periods of interest at the Grignon field site. The shaded areas represent the**
**interquartile range.**

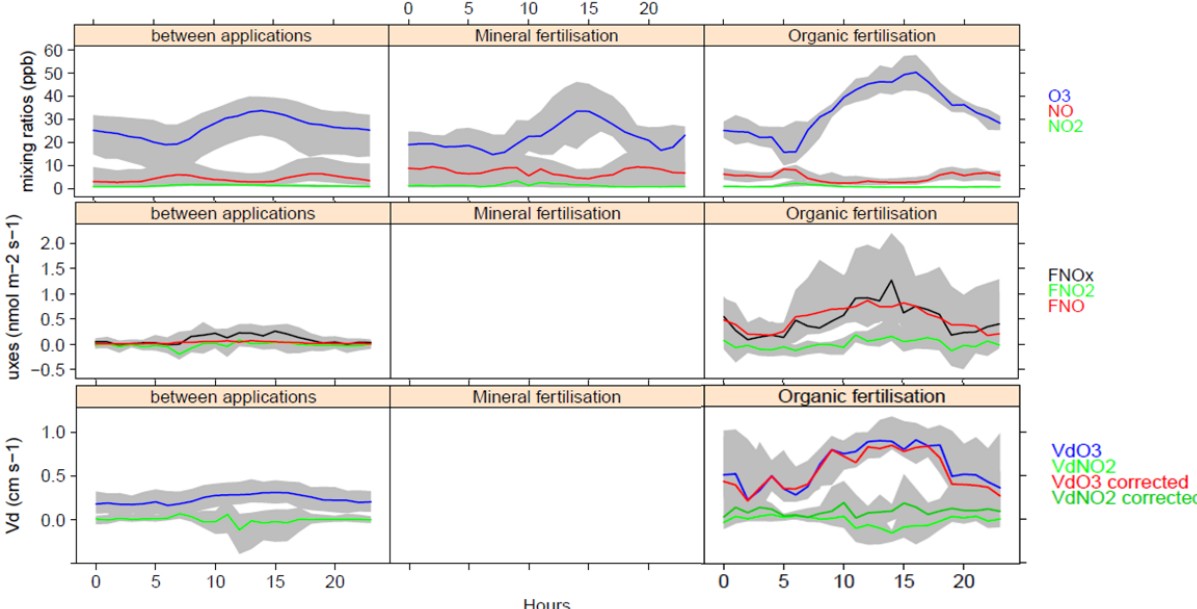

**Figure 7b. Diurnal cycles of NO, NO$_2$ and O$_3$ mixing ratios and fluxes as well as the deposition velocities of NO$_2$ and O$_3$, averaged over the three periods of interest at the Grignon field site. The shaded areas represent the interquartile range. The deposition velocity of NO$_2$ and O$_3$ based on the fluxes accounting for chemical reactions above ground are also shown (VdO3 and VdNO2 corrected).**



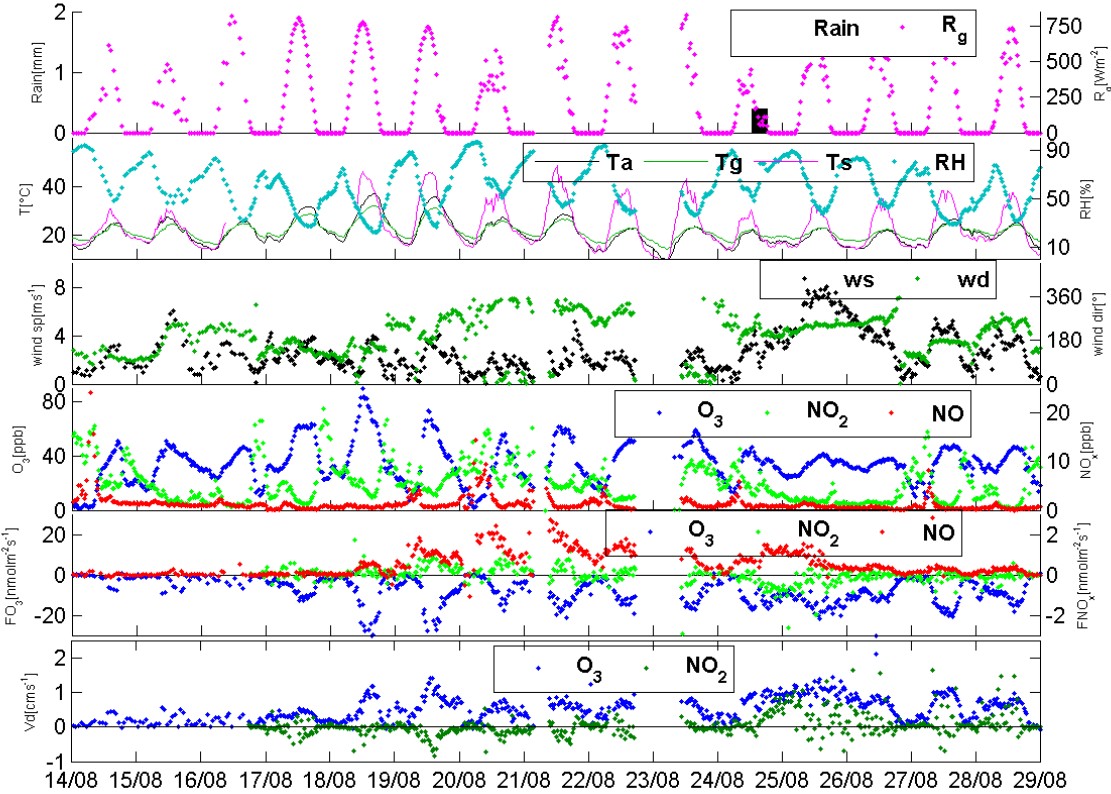

**Figure 8. Meteorological variables and $NO_x$-$O_3$ mixing ratios and fluxes measured during the period 14/08/12 to**
**29/08/12 at the Grignon field site. Ticks on the x-axis correspond to midnight.**


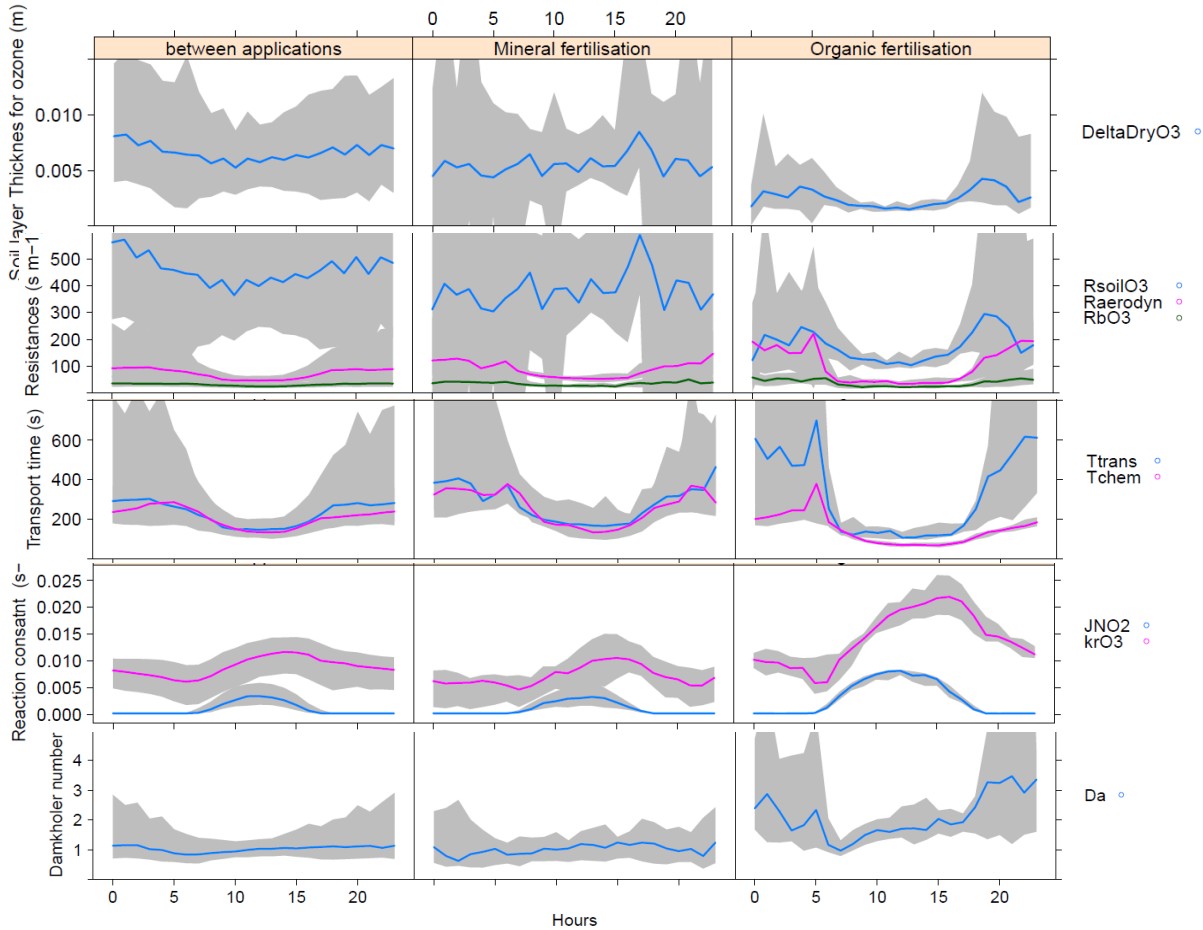

**Figure 9. Diurnal cycles of the $O_3$ penetration depth in the soil (DeltaDryO3), the aerodynamic (Ra), boundary layer**
**(RbO3) and soil resistances (RsoilO3) for $O_3$, the chemical reaction time $\tau_{chem}$ and transport time $\tau_{trans}$, the chemical**
**reaction rates for $NO_2$ photolysis $J_{NO2}$ and NO depletion by $O_3$ ($k_r \times [O_3]$), and the Damköhler number (Da), averaged**
**over the periods of interest at the Grignon field site. The shaded areas represent the interquartile range.**

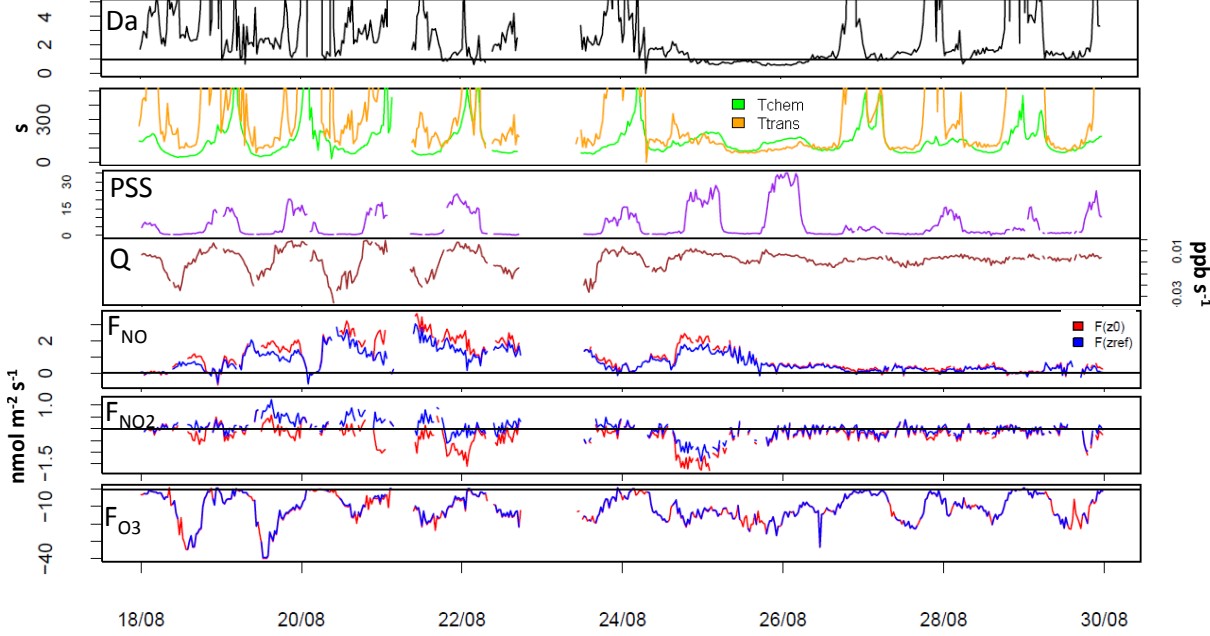

**951 Figure 10. Half-hourly values of photo-stationary state ratio (PSS) and $Q = k_r$ [NO][O$_3$] - $J_{NO2}$[NO$_2$] (s) ; chemistry**
**952 and transport timescales (Tchem and Ttrans) and Damköhler number (Da); measured NO, NO$_2$ and O$_3$ fluxes and**
**953 surface fluxes as computed by assuming a logarithmic flux divergence profile (F$_{NO}$, F$_{NO2}$ and F$_{O3}$) at the Grignon field**
**954 site.**

955

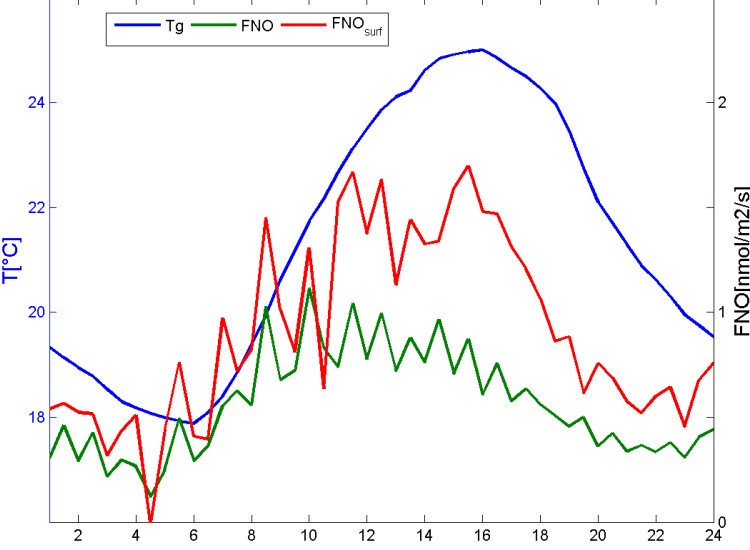

956

**957 Figure 11. Diurnal cycles of ground temperature, NO flux at measurement height and at surface by the logarithmic**

**958 profile.**

959