# Peer review of "Nitrogen oxides and ozone fluxes from an oilseed-rape management cycle: the influence of cattle slurry application"

_Biogeosciences, 2016_

## Referee Comment (RC1) · Anonymous Referee #1 · 3 Aug 2016

I thought this was a very well written article, summarising a study which used the eddy covariance method to monitor variation in NO2+NO (NOx) and O3 in a field of oilseed rape. As the authors point out, there have been few papers which have measured NOx and O3 in the field using this technique and none which have measured them in an agricultural setting. One initial thought it that the article is fairly long, and the editor may feel that some information, particularly in the methods section, may be trimmed. The results and discussion sections are comprehensive and insightful.

I have a limited number of comments/questions listed below.

Line 30 – NOx are toxic to humans above a certain critical limit (http://ec.europa.eu/environment/air/quality/standards.htm) and are also potentially

[Figure]

damaging to ecological systems above a critical load (http://www.apis.ac.uk/indicative-critical-load-values). There are similar, but lower, vales relevant to ozone. This is worth mentioning in the introduction. I think these discussions would help to put your findings in the context of air quality issues. These could be referred to again in the discussion. What do your findings mean in relation to pushing concentrations towards critical limits?

Mention is made of 'significant difference' in the results section but I am unsure of the precise statistical methods used to derive these conclusions. Please add more detail.

This field is surrounded by heavy trafficked roads. To what extent do these roads fall within the flux footprints? Vehicles are prolific producers of NOx and O3. The authors have referred to this in part in the ms, but is it possible to use statistics to unpick the contribution of the field and traffic to the different levels? Perhaps by correlation with traffic densities.

Line 460 - Vehicles also emit VOC which also may affect your interpretation here.

Technical/minor corrections Line 10 – '7-months' should be '7 month' or 'over a period of 7 months' 104 – 'sheep' not 'sheeps'

---

## Referee Comment (RC2) · Anonymous Referee #2 · 16 Aug 2016

The manuscript presents eddy covariance flux measurements for O3, NO and NO2 over an arable field during 8 months between August and the following April. The flux data processing and quality assessment appears sound. The discussion of the NO fluxes (with influence of fertiliser application) and the chemical reactions of the NO-NO2-O3 triad show that the related gas phase chemistry below the flux measurement height can have a considerable effect on the NO and NO2 fluxes but only a very small relative effect on the O3 flux. This finding is in agreement with previous similar studies (e.g. Stella et al., 2012).

However the discussion of the O3 deposition (velocity) and its temporal variation is quite disappointing in its present state. In the discussion of the 8-months time series

of ozone fluxes, the authors mainly concentrate on 2 weeks in August pointing out that the deposition velocity after slurry spreading (average midday values around 0.6 cm/s) was much larger than for the rest of the observation period (average midday values around 0.2 cm/s). They then search for processes that can explain the higher deposition velocity and identify reactive VOC compounds as the most likely cause. I see major shortcomings in this evaluation and interpretation that are detailed in the following comments.

MAJOR COMMENTS

1) A comparison of the observed O3 deposition velocities with previously published data is largely lacking, although it would be very important to compare the magnitude of the values observed in this study with literature results.

2) It is particularly astonishing that the authors do not compare their observations with results for the O3 deposition velocity by Stella et al. (2011) observed over bare soil at the same site. This comparison would reveal that the deposition velocities after slurry application presented here are very similar to previous results over bare soil up to 1 cm/s (also without slurry application). Thus the deposition velocity after slurry spreading are obviously not exceptionally high, but quite normal for bare soil. The need for a special additional chemical sink is therefore not so clear. It rather needs to be discussed why the deposition velocities in the other periods were exceptionally low in the present study, i.e. lower than in other studies over arable/wheat fields (see e.g. Potier et al., 2015, Agric. For. Meteorol.).

3) As shown by Stella et al. (2011) the O3 soil resistance, and thus the deposition velocity, for bare soil strongly depends on the soil surface temperature and/or on the relative surface humidity (for an overview of related processes, see Fowler et al., 2009, Atmospheric Environment).

In Figure 8 (and 7a/b) it can be seen that the strong increase of the dep. velocity at slurry application coincides with a strong increase of the daytime surface temperature

by about 15°C (from <25°C to >40°C). I suggest that the authors apply the relationship for R_soil derived by Stella et al. (2011) to their own dataset, and discuss the deviation of the measurements from that relationship.

4) Figures 5 and 6 are mentioned only shortly in the text and do not provide much additional insight. They could either be omitted or at least reduced (Fig. 5 to one season/whole experiment; Fig. 6 to flux histograms).

5) Title: I am not sure if the title is really appropriate. It makes not much sense to focus on the comparison of "fluxes following an organic and a mineral fertilization" if the first was in high summer and the second in deep winter. It would be more appropriate to mention the observed seasonal/management cycle for the winter crop in the title, and also to discuss it more profoundly.

MINOR COMMENTS

6) Line 160: Please explain what "filtered for outliers" means here. I guess it was a kind of smoothing and gap filling procedure?

7) Line 165: This equation is only valid, if the raw ozone signal has no offset. Why did you not use an analyser sensitivity here (like in the following equations)?

8) Line 166/167: It is unnecessarily confusing to use "s" in these equations twice for two different quantities (in size scaled equations the uppercase and lowercase "s" are often difficult to distinguish). I would recommend to use two different symbols instead for the raw signal and the analyser sensitivity.

9) In Eq. 3 the left side should read "F_NO2" (the subscript 2 is missing in my copy)

10) Line 169: It is not fully appropriate to talk of a "NO2 sensor signal" because there was no NO2 sensor. It would be more correct to name it "NOx sensor signal" or "BLC signal", from which the NO2 flux was derived as a difference to NO (acc. to Eq. 3).

11) Line 171: It is quite confusing to use the same (or a very similar) symbol for the

molar volume and for the deposition velocity in this manuscript. Volumes are generally represented by an uppercase "V". I would suggest to use here the symbol "V_dry".

12) Line 226: This formulation is confusing. Were there two high frequency losses? I think this should be rephrased to "The first main uncertainty was . . ."

13) Line 235f.: How was this high uncertainty effect for NO2 quantified in the uncertainty calculation?

14) Line 298: "which was probably due to a dryer soil in this study." It is not clear which study is meant here. Please make a clearer distinction between "this/these" and "that/those".

15) Lines 346-356: this paragraph is oddly placed here. It should be combined with the text in chapter 3.7.2.

16) Line 401-417: This paragraph with Eq. 10 to 12 should be moved from the discussion to the method section, since this calculation already has been applied previously for the same site (e.g. Stella et al., 2012).

17) Line 420: "Mainly" can be omitted here.

18) Line 448: "is constant" should be omitted here.

19) Line 481f.: If it is assumed that very reactive VOCs significantly contribute to the gas phase destruction of O3 it should also be discussed what the effect of these VOCs on NO could be.

20) Line 770: The unit "N L min-1" is very uncommon. Better use standard liter "sL min-1".

21) Figs. 4, 7, 8, 9: It would be very useful for the reader to use consistent color coding for NO, NO2 and O3 throughout the figures.

22) Fig. 10: The axis labelling is not complete.

---

## Author Comment (AC1) · 22 Sep 2016

**Answer to Anonymous Referee #1**

I thought this was a very well written article, summarising a study which used the eddy covariance method to monitor variation in $NO_2$ + NO ($NO_x$) and $O_3$ in a field of oilseed rape. As the authors point out, there have been few papers which have measured $NO_x$ and $O_3$ in the field using this technique and none which have measured them in an agricultural setting. One initial thought it that the article is fairly long, and the editor may feel that some information, particularly in the methods section, may be trimmed.
We thank referee #1 for this supporting comment. Regarding the methods section, we are open to editor's proposal to move part of it in a section of the supplementary material. We propose to suppress title in line 182 and merge with title in Line 174 as *"Spectral corrections and flux uncertainties"*. We further propose to move sections 2.6 and 2.7 in supplementary material sections S1 and S2 and write a condensed description of these sections below line 189:

*"2.5 Chemical reactions, time scales and flux divergence*

*Chemical reactions between NO, $NO_2$ and $O_3$ are important to consider when interpreting the measured fluxes as they can affect the fluxes above the ground. A common way to determine whether these reactions may indeed affect the flux is through comparison of chemical and transport time scales. Details of the reactions rates, times scales and flux divergence calculations are given in supplementary material sections S1-S3."*

The results and discussion sections are comprehensive and insightful. I have a limited number of comments/questions listed below.

Line 30 – $NO_x$ are toxic to humans above a certain critical limit (http://ec.europa.eu/environment/air/quality/standards.htm) and are also potentially damaging to ecological systems above a critical load (http://www.apis.ac.uk/indicative-critical-load-values). There are similar, but lower, values relevant to ozone. This is worth mentioning in the introduction. I think these discussions would help to put your findings in the context of air quality issues. These could be referred to again in the discussion. What do your findings mean in relation to pushing concentrations towards critical limits?
We thank referee #1 for this comment. We already mentioned toxicity of NOx and O3 at lines 3O and 35. Indeed, $NO_x$ and $O_3$ are toxic to humans and animals. Similarly nitrogen deposition leads to serious adverse effects on vegetation (eutrophication, biodiversity erosion and acidification being the most serious ones), while $O_3$ has a direct adverse effect on plant health through oxidation of photosynthesis pathways and direct tissue destruction above large thresholds. We propose to remove the following text lines 20-31 "$NO_x$, and especially $NO_2$, are toxic gases for humans (WHO, 2013) and national and international authorities regulate their levels", and line 35 *"is a major tropospheric pollutant, harmful for humans and ecosystems, and"*, and add the following text after line 36:
*"$NO_x$, and especially $NO_2$, are toxic gases for humans, increasing risks for various respiratory diseases. World Health Organization gives guidelines for $NO_2$ exposure limits, both annual means (40 $\mu g\ m^{-3}$) and 1-hour mean (200ug/m3) (WHO, 2005: Air quality guidelines for particulate matter, ozone, nitrogen dioxide and sulfur dioxide). For ozone, only a short-term threshold is given (100ug/m3 for 8-hour mean) because there are fewer studies on long-term exposure. These thresholds are established both on epidemiological and toxicological studies on humans and animals. Similarly nitrogen deposition leads to serious adverse effects on vegetation (eutrophication, biodiversity erosion and acidification being the most serious ones), while $O_3$ has a direct adverse effect on plant health through oxidation of photosynthesis pathways and direct tissue destruction above large thresholds. For nitrogen, the concept of critical load has been developed which gives the amount of nitrogen deposition above which an ecosystem is impacted. These critical loads range from 5 kg N $ha^{-1}\ yr^{-1}$ for sensitive habitats to 20 kg N $ha^{-1}\ yr^{-1}$ for less sensitive ones (APIS, 2016). For these reasons, national and international authorities regulate atmospheric levels of these pollutants."*

We propose to also refer to it in the conclusion section as follows (added after line 500):

*"Our findings show that NO emissions from agricultural soils are limited (0.27% of the N-NO applied over the 8 month period, which with a conservative estimation we can extend to a yearly amount). When extended to France with an average nitrogen fertiliser use of 80 kg N $ha^{-1}$ over a fertilised area of around 26 Mha, this*

*would lead to emission of $NO_x$ of around ~5.6 t N-NO $ha^{-1}$, which is negligible compared to transport and industry which is several hundreds of thousand larger (CITEPA, 2015). The seasonality of these emissions may however lead to air quality issues during spring and late summer-autumn which are the main fertiliser application periods. Indeed, most of the emission we measured occurred with a few weeks following fertilisation. In terms of ozone, our findings, and previous ones, show that ozone is efficiently deposited throughout the year. This means that crops are participating through this process in the reduction of the atmospheric oxidising capacity"*

Answers to referee #2 provide more details on $O_3$ deposition analysis.

Mention is made of 'significant difference' in the results section but I am unsure of the precise statistical methods used to derive these conclusions. Please add more detail.
We propose to explain what was meant by 'significant' each time the term is used in the text. The term was misused in line 336 where we propose to replace it by *"The $NO_2$ flux daily pattern was different"*. Otherwise, we used Student t-tests to check a difference in mean.

This field is surrounded by heavy trafficked roads. To what extent do these roads fall within the flux footprints? Vehicles are prolific producers of $NO_x$ and $O_3$. The authors have referred to this in part in the ms, but is it possible to use statistics to unpick the contribution of the field and traffic to the different levels? Perhaps by correlation with traffic densities.
This is a good question indeed. The field is surrounded by heavy traffic roads. Unfortunately we do not have traffic statistics at that location for the given period. To answer this question, we propose to evaluate the footprint of the roads using the FIDES flux and concentration footprint model (Loubet et al., 2010), which is essentially similar to the Korman & Meixner model (Kormann and Meixner, 2001) but with a different treatment of the lateral dispersion. Overall the flux footprint from the nearby roads is smaller than 1% (which means that only 1% of the roads emission contributes to the flux at the mast) most of the time, but the concentration footprint reaches up to 10% during some episodes, with different roads contributing differently during different periods.

[Figure]

**Figure R1. Flux and concentration footprints of the field and surrounding roads calculated with the FIDES model.**

Using the flux and concentration footprint allows evaluating the contribution of traffic to the $NO_x$ concentration and fluxes. For that, a conservative emission of 250 mg $km^{-1}$ $vehicle^{-1}$ was considered. The average vehicle count per day ranges from 5000 to 13000 on the surrounding roads (2010 counts, "Statistiques du département des Yvelines pour 2010"). Using an average of 10000 vehicles per day, we can calculate that the flux due to the surrounding roads may be of magnitude 4% to 40% of the measured flux. However, the $NO_x$ vehicles emissions have a sporadic nature: indeed, 10000 vehicles per day means a maximum of 1 vehicle every ~2 second if we consider, conservatively, that most of the traffic is condensed during 9 hours only. This vehicle is moving at say 90 km $h^{-1}$ (25 m s-1) hence leading to a moving point source of $NO_x$. We therefore expect that the signal of this moving and sporadic source is not captured by the eddy-covariance method, and would be filtered out by despiking and flux calculations (Foken, 2008). From another perspective, we cannot exclude that we have not biased the flux by filtering out the flux coming from the roads as discussed by Mahrt (2010)

We propose to add the following discussion section after line 316:

*Using the FIDES flux and concentration footprint model (Loubet et al., 2010) we evaluated the footprint of nearby roads. Overall the flux footprint from the nearby roads was smaller than 1% (which means that only 1% of the roads emission contributes to the flux at the mast) most of the time, but the concentration footprint reaches up to 10% during some episodes, with different roads contributing differently during different periods (Figure S1). Assuming a conservative emission of 250 mg km$^{-1}$ vehicle$^{-1}$ and an average vehicle count 10000 vehicles per day (2010 counts, "Statistiques du département des Yvelines pour 2010" shows range between 5000 and 15000), we evaluate a contribution from 4% to 40% of the road on the measured flux. However, since vehicles emissions of NO$_x$ have a sporadic nature. Indeed 10000 vehicles per day means a maximum of 1 vehicle every ~2 second (if we consider, conservatively, that most of the traffic is condensed during 9 hours only). These vehicles are also moving at say 90 km h$^{-1}$ (25 m s$^{-1}$) hence leading to a moving point source of NO$_x$. We therefore expect that the signal of this moving and sporadic source is not captured by the eddy-covariance method, and would be filtered out by despiking and flux calculations (Foken, 2008). From another perspective, we cannot exclude that we have not biased the flux by filtering out the flux coming from the roads as discussed by Mahrt (2010)."*

We further propose to include Figure R1 in Supplementary section S4 and add following text to introduce the figure:

*"S4 Flux and concentration footprint*

*The flux and concentration footprint was roughly estimated for each of the major roads around the site. Each road was geo-localised and assumed 10 m width. The FIDES model was computed with field roughness (z$_0$), friction velocity (u∗) and Obukhov length (L)."*

Line 460 - Vehicles also emit VOC which also may affect your interpretation here.
This is true indeed, but the amount of VOC to NO$_x$ emitted is small (below 1% for non methanic VOC, according to French national emissions inventory, CITEPA (2015), not considering CO) and hence the effect on O$_3$ concentrations and flux is expected to be of second order. Moreover, we see increased O$_3$ deposition on a period (August) which has the lowest traffic density throughout the year in this area, and we do not see increased deposition during high traffic load days throughout the year, which points towards a small effect of VOC emitted by vehicles on the O3 flux in August.
However, we have tempered our interpretation on the potential effect of VOC emitted by slurry on ozone flux since, as pointed out by the reviewer #2, the increase of ozone following fertilisation may have other physical and chemical explanations.

**Technical/minor corrections**
Line 10 – '7-months' should be '7 month' or 'over a period
of 7 months'
Thanks for this suggestion. We propose to correct for it.

L104 – 'sheep' not 'sheeps'
Thanks for this suggestion. We propose to correct for it.

**References**
Foken, T.: The energy balance closure problem: An overview, Ecological Applications, 18, 1351-1367, 2008.
Kormann, R. and Meixner, F. X.: An analytical footprint model for non-neutral stratification, Boundary Layer Meteorol., 99, 207-224, 2001.
Loubet, B., Genermont, S., Ferrara, R., Bedos, G., Decuq, G., Personne, E., Fanucci, O., Durand, B., Rana, G., and Cellier, P.: An inverse model to estimate ammonia emissions from fields, European Journal of Soil Science, 61, 793-805, 2010.
Mahrt, L.: Computing turbulent fluxes near the surface: Needed improvements, Agric. For. Meteorol., 150, 501-509, 2010.

---

## Author Comment (AC2) · 22 Sep 2016

**Answer to Anonymous Referee #2**

The manuscript presents eddy covariance flux measurements for O3, NO and NO2 over an arable field during 8 months between August and the following April. The flux data processing and quality assessment appears sound. The discussion of the NO fluxes (with influence of fertiliser application) and the chemical reactions of the NONO2-O3 triad show that the related gas phase chemistry below the flux measurement height can have a considerable effect on the NO and NO2 fluxes but only a very small relative effect on the O3 flux. This finding is in agreement with previous similar studies (e.g. Stella et al., 2012).

However the discussion of the O3 deposition (velocity) and its temporal variation is quite disappointing in its present state. In the discussion of the 8-month time series of ozone fluxes, the authors mainly concentrate on 2 weeks in August pointing out that the deposition velocity after slurry spreading (average midday values around 0.6 cm/s) was much larger than for the rest of the observation period (average midday values around 0.2 cm/s). They then search for processes that can explain the higher deposition velocity and identify reactive VOC compounds as the most likely cause. I see major shortcomings in this evaluation and interpretation that are detailed in the following comments.

We thank Referee #2 for this comment. We have put more emphasis on NO and $NO_2$ than $O_3$ in this manuscript because, as spotted by referee # 2, we have already published a lot of results on $O_3$ on this site while NO and $NO_2$ flux measurements are newer. We nevertheless propose to include a result and discussion section specifically on ozone fluxes and deposition velocity. We agree that we might have put too much emphasis on the potential reaction between VOC compounds and $O_3$ in the current manuscript and therefore propose to compare our $O_3$ deposition velocity with existing literature and especially with the parameterisation we had on bare soil in Stella et al. (2011).

**MAJOR COMMENTS**
1) A comparison of the observed $O_3$ deposition velocities with previously published data is largely lacking, although it would be very important to compare the magnitude of the values observed in this study with literature results.
This is indeed a good comment. We have included in the Figure R2 below the ozone deposition velocity during the period as averaged daily patterns. We propose to include Figure R2 and the following short paragraph in a supplementary material S6 section on ozone seasonal pattern:

*S6. Seasonal pattern of $O_3$ deposition velocity and NO fluxes*

*We found similar magnitude of ozone fluxes in August and September as those reported by Stella et al. (2013) over a meadow during the summer. We also found similar nocturnal $O_3$ deposition velocity as found by Stella et al. (2011) over bare soil during summer, but higher daily maximum (0.8 cm s-1 instead of 0.5-0.6 cm s-1). We further find a similar midday magnitude as Stella et al. (2011) found in April with wetter soils. Night-time ozone deposition velocity does not go lower than around 0.2 cm $s^{-1}$ in our study, as also found by Zhu et al. (2015) over a growing wheat in China, Stella et al. (2011) over bare soil in summer, and Lamaud et al. (2009) over maize. These authors as well as Huang et al. (2016) clearly show that this is due to non-stomatal deposition being primarily driven by $u_*$ which does not reach zero at night during these periods. We can hence conclude that we found consistent ozone deposition in August and September compared to other studies at that site or in other geographical areas. When compared to previous years on the same site the deposition velocity measured during the winter in this study was clearly smaller. We interpret this as being primarily due to $u_*$ being smaller that winter compared to other winters as well as due to a slow growth of the winter crop due to soil drought in September (SWC =20% in the 15 cm horizon).*

[Figure]

**Figure R2. Seasonal changes of ozone deposition velocity VdO3 and NO fluxes. Lines show median and grey area inter-quantiles.**

2) It is particularly astonishing that the authors do not compare their observations with results for the O3 deposition velocity by Stella et al. (2011) observed over bare soil at the same site. This comparison would reveal that the deposition velocities after slurry application presented here are very similar to previous results over bare soil up to 1 cm/s (also without slurry application). Thus the deposition velocity after slurry spreading is obviously not exceptionally high, but quite normal for bare soil. The need for a special additional chemical sink is therefore not so clear. It rather needs to be discussed why the deposition velocities in the other periods were exceptionally low in the present study, i.e. lower than in other studies over arable/wheat fields (see e.g. Potier et al., 2015, Agric. For. Meteorol.).

As just exposed in the answer to question 1) of Referee #2, we found similar night time deposition velocities as in Stella et al. (2011) but larger daytime values during the summer and similar as in April 2007, which showed also wetter soils. Regarding the other periods, the fact that we report smaller ozone deposition velocity than in Potier et al. (2015) is due to the periods which are spanned in this study: from August to March, which correspond to the period with the lowest stomatal component and also the smallest non-stomatal component because of the small leaf area index of the rapeseed. This winter also showed an especially low $u_*$.

3) As shown by Stella et al. (2011) the O3 soil resistance, and thus the deposition velocity, for bare soil strongly depends on the soil surface temperature and/or on the relative surface humidity (for an overview of related processes, see Fowler et al., 2009, Atmospheric Environment). In Figure 8 (and 7a/b) it can be seen that the strong increase of the dep. velocity at slurry application coincides with a strong increase of the daytime surface temperature by about 15_C (from <25_C to >40_C). I suggest that the authors apply the relationship for R_soil derived by Stella et al. (2011) to their own dataset, and discuss the deviation of the measurements from that relationship.

This is a very good suggestion which we followed. We propose to add Figure R3 and R4 together with the following paragraph in the supplementary material S7 section on ozone seasonal pattern:

*S7. Comparison of ozone fluxes to Stella et al. (2011)*

*In order to compare to previous studies of ozone deposition to bare soil on the same site, we have calculated the soil surface resistance as in Stella et al. (2011) and deduced the ozone deposition velocity as $V_{dO3} = (R_{soilO3} + R_{bO3} + R_a(z_{ref}))^{-1}$. In this way, we can compare the two studies while excluding any confounding factors (roughness and turbulent exchange intensity). We can see in Figure (R3) that the measured ozone deposition velocity during August follows most of the time the parameterisation of Stella et al. (2011) except for some days including 18 and 19 August which corresponds to slurry application and 24, 25, 26 August, which follows a small rainfall. We also see an overestimation of the Stella parameterisation before the18 August, which we interpret as being due to the straw and wheat residues laying on the ground before slurry incorporation. This comparison hence demonstrates that the ozone deposition was indeed increased slightly following slurry application and subsequently following rainfall. This may be either due to a physical reason (increased surface exchange on the soil due to tillage or humidity change due to slurry) or a chemical reason (surface reactivity*

*changes due to added organic matter or VOC emissions from slurry). Figure R4 further shows that the main differences are observed for wet soils and relatively low temperatures (this is after rainfall) and to a lesser extent for dryer and hotter situation (following slurry spreading).*

[Figure]

**Figure R3. Comparison of ozone deposition velocity from this study (black dots), and from the parameterisation of Stella et al. (2011) (red line) based on surface temperature.**

[Figure]

**Figure R4. Response of ozone deposition velocity to surface humidity RH($z_0$) and surface temperature T($z_0$). Shown are data from this study and from the parameterisation of Stella et al. (2011). Period from 14 August to 6 September which is before and after slurry spreading and corresponds to Figure R3**

4) Figures 5 and 6 are mentioned only shortly in the text and do not provide much additional insight. They could either be omitted or at least reduced (Fig. 5 to one season/whole experiment; Fig. 6 to flux histograms). That sounds like a fair suggestion. However we feel that these are still important and therefore propose to put these two figures in a supplementary material section S5, and refer to in the text. *"S5 Wind roses and histograms of NO, NO2 and O3 concentration at the site."*

5) Title: I am not sure if the title is really appropriate. It makes not much sense to focus on the comparison of "fluxes following an organic and a mineral fertilization" if the first was in high summer and the second in deep

winter. It would be more appropriate to mention the observed seasonal/management cycle for the winter crop in the title, and also to discuss it more profoundly.

*This is a sound suggestion. We propose to change the title to: "Nitrogen oxides and ozone fluxes above oilseed-rape with emphasis on organic fertilisation"*

**MINOR COMMENTS**

6) Line 160: Please explain what "filtered for outliers" means here. I guess it was a kind of smoothing and gap filling procedure?

Points away from median lag ± standard deviation were considered as outliers for the lag.

7) Line 165: This equation is only valid, if the raw ozone signal has no offset. Why did you not use an analyser sensitivity here (like in the following equations)?

Thank you for this comment. Indeed, the ratio method as described in Muller et al. (2010) and as applied here, considers that the signal has no offset. The alternative methods, computation of the flux by use of sensitivity and offset values, fixes this problem but, on the other hand, present shortcoming in terms of offset and sensitivity determination. Indeed, this kind of fast-sensor make use of cumarine disks whose sensitivity decrease on a weekly timescale, but this is not taken into account in the sensitivity estimation by signal regression or by disk calibration, that assumes sensitivity to be constant for each disk. We chose the ratio method that neglects offset but takes into account the sensitivity variation along the lifetime of the cumarine disks. An example of regression between fast- and slow- sensors signals for a three-day period is given in fig. R5. With a negative offset of 0.04V, the relative error on the flux, that can be estimated as the ratio between the offset and the raw signal, would range between about 4% to 20% (for ambient concentration of 80 and 20ppb respectively).

[Figure]

**Figure R5. O3 fast sensor signal versus slow-response analyser signal. Half-hour means for a period of 3 days.**

8) Line 166/167: It is unnecessarily confusing to use "s" in these equations twice for two different quantities (in size scaled equations the uppercase and lowercase "s" are often difficult to distinguish). I would recommend to use two different symbols instead for the raw signal and the analyser sensitivity.

We propose to use now the name of the compound instead of *s* for signals.

9) In Eq. 3 the left side should read "F_NO2" (the subscript 2 is missing in my copy)

Thanks for spotting this. We propose to change for $F_{NO2}$.

10) Line 169: It is not fully appropriate to talk of a "NO2 sensor signal" because there was no NO2 sensor. It would be more correct to name it "NOx sensor signal" or "BLCsignal", from which the NO2 flux was derived as a difference to NO (acc. to Eq. 3).

Thanks for this suggestion. We propose to use BLC signal.

11) Line 171: It is quite confusing to use the same (or a very similar) symbol for the molar volume and for the deposition velocity in this manuscript. Volumes are generally represented by an uppercase "V". I would suggest to use here the symbol "V_dry".

Thanks for this suggestion. We propose to use $v_{dry}$ instead. Overall we propose to change text and equations frm lines 165 to 173 as:.

$$F_{O3} = \frac{\overline{\chi_{O3}}}{V_{dry}} \frac{\overline{w'O_3'}}{\overline{O_3}} \tag{1}$$

$$F_{NO} = \frac{\overline{w'NO'}}{S_{NO}V_{dry}} \tag{2}$$

$$F_{NO} = \frac{1}{\alpha V_{dry}} \left( \frac{\overline{w'NO_x'}}{S_{NO2}} - \frac{\overline{w'NO'}}{S_{NO}} \right) \tag{3}$$

*where $O_3$ (in mV), NO and $NO_x$ (in counts $s^{-1}$) are the uncalibrated fast signals, $\chi_{O3}$ is the 30 min average of the slow-sensor reference $O_3$ mixing ratio (in ppb), while $S_{NO}$ and $S_{NO2}$ are the sensitivity of the analysers (in counts $s^{-1}$ $ppb^{-1}$). $\alpha$ is the blue light converter conversion efficiency, and $V_{dry}$ is the molar volume of dry air (in $m^3$ $mol^{-1}$). All fluxes (momentum, heat, $CO_2$, $H_2O$, NO, $NO_2$, $O_3$) were computed by the Eddypro software and final flux data were averaged for 30 min intervals. Evaluation methodologies from the CarboEurope project were applied - see (Aubinet et al., 2000 ; Loubet et al., 2011).*

12) Line 226: This formulation is confusing. Were there two high frequency losses? I think this should be rephrased to "The first main uncertainty was : : :"
Indeed this formulation was misleading. We propose to rephrase with *"The largest uncertainty ..."*. We then rephrase Line 220 as *"The second largest uncertainty ..."*

13) Line 235f.: How was this high uncertainty effect for NO2 quantified in the uncertainty calculation?
Actually, we have mistaken here $NO_2$ for $NO_x$. As the uncertainty analysis was performed on the $NO_x$ channel. Hence we have changed the text accordingly: *"A higher random uncertainty was found for NOx fluxes which were smaller than NO fluxes and with a relatively low conversion ratio from NO2 to NO (30%)"*

14) Line 298: "which was probably due to a dryer soil in this study." It is not clear which study is meant here. Please make a clearer distinction between "this/these" and "that/those".
We propose to change for: *". Stella et al. (2012) measured larger peak of NO emissions following slurry spreading, but only lasting two to three days, which was probably due to a dryer soil in our study compared to Stella et al. (2012). "*

15) Lines 346-356: this paragraph is oddly placed here. It should be combined with the text in chapter 3.7.2.
This is a sound suggestion we propose to move and cut this paragraph to include section 3.7.2 as follows:
*"Reactive VOCs such as sesquiterpenes and monoterpenes were previously found to be emitted from soils (Horvath et al., 2012; Penuelas et al., 2014), and some of these sesquiterpenes species react with $O_3$ in the order of a few seconds. The reactions of $O_3$ with larger terpenes are important sources of OH, as well as the ozonolysis of simpler unsaturated compounds. (Donahue et al., 2005). "*

16) Line 401-417: This paragraph with Eq. 10 to 12 should be moved from the discussion to the method section, since this calculation already has been applied previously for the same site (e.g. Stella et al., 2012).
This is indeed a good suggestion. We propose to move lines 402-417 to a supplementary section after the chemical time scales (actual sections 2.7):

*S3 Evaluating the flux difference between ground and the reference height*

*When chemical timescale is shorter than transport timescale, chemical reactions affect concentrations and fluxes, resulting in flux divergence. This causes the flux at the measurement point to be different from the surface flux. The flux difference can be evaluated with a method developed by Duyzer et al. (1995) based on the early developments of Lenschow (1982) and Lenschow and Delany (1987). This method assumes a logarithmic profile of the flux divergence and depends on measured mixing ratios, stability function and friction velocity:*

$$(\partial F / \partial z)_z = a \ln z + b \tag{10}$$

$$a_{NO_2} = -a_{NO} = -a_{O_3} = -\Phi_H / ku_* \left[ k_r \left( \overline{[NO]} F_{O_3, z_{ref}} + \overline{[O_3]} F_{NO, z_{ref}} \right) - j_{NO_2} F_{NO_2, z_{ref}} \right] \tag{11}$$

*Here $\overline{[NO]}$ and $\overline{[O_3]}$ are mixing ratios which should ideally refer to the geometric mean height of the profile measurements but was taken from the measurement height in our study, $z_{ref}$ is the measurement height and $\Phi_H$ is the stability correction function for heat estimated at $z_{ref}$ (Dyer and Hicks, 1970). Coefficient b of Eq. 10 can be computed as $b = -a \ln(z_2)$ where $z_2$ is the height above which the flux divergence is zero. Duyzer et al. (1995) showed with numerical simulations that the $NO_x$ flux divergence could be approximated by Eq. 10 below a height of 4m, while it was negligible above. We refer to 4 m as the reference height $z_2$ at which we assume the*

*flux divergence to be zero. Equation 10 can be integrated from measurement height to any height, for each compound giving:*

$$F(z_{z0}) = F(z_{ref}) + a(1 + ln(z_2))(z_{ref} - z_0) - a[z_{ref}ln(z_{ref}) - z_0 ln(z_0)] \hspace{2cm} (12)$$

We propose to replace *"Mainly due"* at start of Line 420 by *We quantified this variation by numerically solving Eq. 12, based on the model of Duyzer et al. (1995). Due*

17) Line 420: "Mainly" can be omitted here.
We propose to delete.

18) Line 448: "is constant" should be omitted here.
We propose to delete.

19) Line 481f.: If it is assumed that very reactive VOCs significantly contribute to the gas phase destruction of O3 it should also be discussed what the effect of these VOCs on NO could be.
This is a sound remark indeed. VOCs are known to form intermediates $R\dot{O}_2$ and $HO_2$ radicals which react with NO, converting NO to $NO_2$ (Atkinson, 2000). Once VOCs are emitted, they are broken down chemically into free radicals. The degradation reactions of VOCs lead to the formation of intermediate $RO_2$ and $HO_2$ radicals that can further react with NO, converting NO to $NO_2$ which, after photolysis, form $O_3$.

$HO_2 + NO \rightarrow OH + NO_2$

$RO\cdot_2 + NO \rightarrow RO\cdot + NO_2$

The ozone formation chain is then determined by the competition between the peroxy radicals ($HO_2$) reactions and NO and the peroxy radical termination reactions. The factors that affect the number of NO molecules converted to $NO_2$ will also affect the rate of $O_3$ formation. These factors include radical sources and sinks, $NO_x$ sinks and reaction pathways of NO molecules converted to $NO_2$ in the VOC's degradation mechanism. The photochemical formation of $O_3$ vs. photochemical loss of $O_3$ in the troposphere depends therefore on the NO concentration and is determined by the rate of the reaction of the $HO_2$ radicals with NO. (Atkinson, 2007; Monks, 2005).

20) Line 770: The unit "N L min-1" is very uncommon. Better use standard liter "sL min-1".
We have several example of NL min-1. However if editor suggest better using sL we propose to change for sL.

21) Figs. 4, 7, 8, 9: It would be very useful for the reader to use consistent color coding for NO, $NO_2$ and $O_3$ throughout the figures.
We propose to set all colors as blue for $O_3$, green for $NO_2$, and red for NO. We have hence changed figure 7b to the following:

[Figure]

**Figure 7b. Diurnal cycles of NO, NO₂ and O₃ mixing ratios and fluxes as well as the deposition velocities of NO₂ and O₃, averaged over the three periods of interest at the Grignon field site. The shaded areas represent the interquartile range.**

22) Fig. 10: The axis labelling is not complete.

Indeed, although Da and PSS has no units, Q units are missing. We propose to add labels (ppb s⁻¹). The new figure 10 looks like this:

[Figure]

References

Atkinson, R.: Atmospheric chemistry of VOCs and NOx, Atmos. Environ., 34, 2063-2101, 2000.

Atkinson, R.: Rate constants for the atmospheric reactions of alkoxy radicals: An updated estimation method, Atmos. Environ., 41, 8468-8485, 2007.

Aubinet, M., Grelle, A., Ibrom, A., Rannik, U., Moncrieff, J., Foken, T., Kowalski, A. S., Martin, P. H., Berbigier, P., Bernhofer, C., Clement, R., Elbers, J., Granier, A., Grunwald, T., Morgenstern, K.,

Pilegaard, K., Rebmann, C., Snijders, W., Valentini, R., and Vesala, T.: Estimates of the annual net carbon and water exchange of forests: the EUROFLUX methodology, Advances in Ecological Research, 30, 113-175, 2000.

Donahue, N. M., Hartz, K. E. H., Chuong, B., Presto, A. A., Stanier, C. O., Rosenhorn, T., Robinson, A. L., and Pandis, S. N.: Critical factors determining the variation in SOA yields from terpene ozonolysis: A combined experimental and computational study, Faraday Discussions, 130, 295-309, 2005.

Duyzer, J. H., Deinum, G., and Baak, J.: The Interpretation of Measurements of Surface Exchange of Nitrogen-Oxides - Correction for Chemical-Reactions, Philos T R Soc A, 351, 231-248, 1995.

Dyer, A. J. and Hicks, B. B.: Flux-profile relationship in the constant flux layer, Q. J. Roy. Meteor. Soc., 96, 715-721, 1970.

Horvath, E., Hoffer, A., Sebok, F., Dobolyi, C., Szoboszlay, S., Kriszt, B., and Gelencser, A.: Experimental evidence for direct sesquiterpene emission from soils, J Geophys Res-Atmos, 117, 2012.

Huang, L., McDonald-Buller, E. C., McGaughey, G., Kimura, Y., and Allen, D. T.: The impact of drought on ozone dry deposition over eastern Texas, Atmos. Environ., 127, 176-186, 2016.

Lamaud, E., Loubet, B., Irvine, M., Stella, P., Personne, E., and Cellier, P.: Partitioning of ozone deposition over a developed maize crop between stomatal and non-stomatal uptakes, using eddy-covariance flux measurements and modelling, Agric. For. Meteorol., 149, 1385-1396, 2009.

Lenschow, D. and Delany, A. C.: An analytic formulation for NO and NO2 flux profiles in the atmospheric surface layer, J. Atmos. Chemis., 5, 301-309, 1987.

Lenschow, D. H.: Reactive trace species in the boundary layer from a micrometeorological perspective, J. Meteorol. Soc. Jpn., 60, 472-480, 1982.

Loubet, B., Laville, P., Lehuger, S., Larmanou, E., Flechard, C., Mascher, N., Génermont, S., Roche, R., Ferrara, R. M., Stella, P., Personne, E., Durand, B., Decuq, C., Flura, D., Masson, S., Fanucci, O., Rampon, J.-N., Siemens, J., Kindler, R., Schrumpf, M., Gabriele, B., and Cellier, P.: Carbon, nitrogen and Greenhouse gases budgets over a four years crop rotation In northern France, Plant and Soil, 343, 109-137, 2011.

Monks, P. S.: Gas-phase radical chemistry in the troposphere, Chemical Society Reviews, 34, 376-395, 2005.

Penuelas, J., Asensio, D., Tholl, D., Wenke, K., Rosenkranz, M., Piechulla, B., and Schnitzler, J. P.: Biogenic volatile emissions from the soil, Plant Cell and Environment, 37, 1866-1891, 2014.

Potier, E., Ogee, J., Jouanguy, J., Lamaud, E., Stella, P., Personne, E., Durand, B., Mascher, N., and Loubet, B.: Multi layer modelling of ozone fluxes on winter wheat reveals large deposition on wet senescing leaves, Agric. For. Meteorol., 211, 58-71, 2015.

Stella, P., Kortner, M., Ammann, C., Foken, T., Meixner, F. X., and Trebs, I.: Measurements of nitrogen oxides and ozone fluxes by eddy covariance at a meadow: evidence for an internal leaf resistance to NO2, Biogeosciences, 10, 5997-6017, 2013.

Stella, P., Loubet, B., Lamaud, E., Laville, P., and Cellier, P.: Ozone deposition onto bare soil: a new parameterisation, Agric. For. Meteorol., 151, 669-681, 2011.

Stella, P., Loubet, B., Laville, P., Lamaud, E., Cazaunau, M., Laufs, S., Bernard, F., Grosselin, B., Mascher, N., Kurtenbach, R., Mellouki, A., Kleffmann, J., and Cellier, P.: Comparison of methods for the determination of NO-O-3-NO2 fluxes and chemical interactions over a bare soil, Atmos. Meas. Tech., 5, 1241-1257, 2012.

Zhu, Z. L., Sun, X. M., Zhao, F. H., and Meixner, F. X.: Ozone concentrations, flux and potential effect on yield during wheat growth in the Northwest-Shandong Plain of China, J. Environ. Sci., 34, 1-9, 2015.

CITEPA, 2015. Inventaire des émissions de polluants atmosphériques et de gaz à effet de serre en France – Format SECTEN. © CITEPA 2015.

APIS, 2016. http://www.apis.ac.uk/indicative-critical-load-values

---

## Referee Report (RR1)

The manuscript titled 'Nitrogen oxides and ozone fluxes for an oilseed-rape management: influence of organic fertilisation' by Vuolo et al. discusses fluxes of NO, $NO_2$ and $O_3$ measured using the eddy covariance method. The paper is well written and structured and describes an interesting dataset which I believe the flux community will like to see. I recommend that the paper should be published subject to some small edits which I leave to the editors and author's discretion. I hope that my comments aid the authors and look forward to seeing the paper published.

**Comments:**

It may be prudent to change the title to 'Nitrogen oxides and ozone fluxes from an oilseed-rape management cycle: the influence of cattle slurry application'. The term organic fertilisation covers a wide variety of possibilities and as the authors point out, much of the chemistry occurring can be dominated by VOC emissions which will vary widely depending on fertiliser type and consistency.

As mentioned in the previous review of the paper, it is a fairly long submission. The author's attempts to shorten the paper by converting some of the methodology section to supplementary material do help with this. I don't believe that shortening the paper further would improve its readability and would only serve to damage its scientific value.

The soil pH is relatively high (7.6). Is this normal for the field or due to recent liming? There is little mention of this on nitrification rates.

There is mention of the FIDES footprint analysis which shows that the effect of pollution from the cars would be minimal, but no graphical representation of this. Would it be possible to include a rough sketch of the field site and location of the roads with a representation of the footprint contribution during the measurement period?

What percentage of eddy covariance measurements passed QC steps for each compound? What is the total time coverage for each?

Which version of eddy pro was used? Were any other settings changed in eddypro outside of the carboEurope settings to accommodate NO, $NO_2$, $O_3$. (i.e. spike removal, outliers etc…) If so please include a brief summary.

L190: It would be useful to give the reader a range of Reynolds number that would be in the turbulent range for the site in brackets. i.e (xxxx to xxxx)

I would like to see a bit more detail in how the fluxes were quality controlled. What were the cut-off values for outliers and why? Was u* limited used as a cut-off? If so, what limit was chosen and why? If not, why not?

L213: At what point would the authors deem the eddy covariance method unusable?

Section 3.3.1: These observations seem worrying when applying eddy covariance to such reactive compounds which are constantly changing as they disperse from sources. Later in Section 3.7 an estimate of 4 to 40% of contribution to fluxes is described based solely on estimated stats. Horizontal transfer of the species being measured in and out of the storage

area of the fetch and the resultant advection effect seems to be a very significant source of uncertainty and one which cannot be fully accounted for in this study even with de-spiking etc... It is understandable that no field site is perfect, but as one of the aims of this study is to asses if eddy covariance is suitable for the measurements it seems odd to accept such a large source of error as a given. Perhaps some of the more negative aspects of the methodology should be embraced as a discussion point for future studies?

Is it possible to report a detection limit for each of the measured fluxes?

Perhaps this comment is beyond the scope of the paper and I do not expect the authors to amend the manuscript. Was short term changes in PAR (i.e. the effect of clouds) compared to the random error of Ozone fluxes? Is this not an issue when looking at fluxes over a 30 min period when UV exposure can change so dramatically over very short time periods? If fluxes were calculated over a 60 min period instead of 30 minutes are the same fluxes and correlations observed? Would it reduce cumulative flux uncertainty at the cost of data points and the observation of diurnal patterns?

Section 3.6: How were cumulative totals estimated? Linear regression between points or using the diurnal cycles to gap fill? With such consistent patterns and correlations it seems like gap filling could be modelled relatively well?

Axis Text on Fig 7 has overlapped in places

**Optional:**

I don't like the phrase 'changed sign' referring to fluxes switching between emission and uptake at different levels. L23 & L 526**.** If possible please re-word.

L21: replace 'at all times during' with 'constantly throughout'

L 37: replace 'increasing risks for' with 'exposure to which increases risk to'

L48: replace 'mostly due to' with 'primarily the by-products of'

L136: replace ~monthly with 'approximately once a month'

L193*This is assumed to be "white noise" and……

L241: replace 'on' with 'over'

L246: replace 'strongest' with 'highest'

L258: …it can be deduced that deposition velocities were around…

L337: probably similar to those measured in September

**Technical:**

Indents throughout the manuscript are inconsistent. Editorial team will correct?

Dates are presented inconsistently throughout. Choose either (18th of February) or (18/02/16) format and stick with it.

L131: *December

L124 * performed on

L223: *7 month period

L25: replace 'than' with 'of'

L297: 0.27 or 0.25? See line295

L330*24th

L397: delete 'of'

---

## Author Response (AR3)

**Answer to Anonymous Referee #1**

I think this is an excellent study, with the MS including particularly detailed discussion of the underlying processes which have driven the fluxes measured. I think this adds to our knowledge of NO, $NO_2$ and $O_3$ fluxes and would be of interest to the readership of Biogeosciences. Whilst I find the scientific updates to have been well implemented in general, for example I am impressed by the detail given for the response to the influence of the surrounding roads (3.7) I think that some minor changes will still be required before publication. There is still no mention of the statistics in the methods section. I think that this is essential if the reader is to make an informed decision about the drivers of flux increases. What has been tested against what? What was the magnitude of change and what is the statistical output?

The different measurements and their relationship are analysed in sections 3.3 to 3.8. In particular, we focus on the relationship between $NO_x$ and $O_3$ fluxes and meteorological and soil parameters (sections 3.3.2 and 3.3.4), nitrogen input (section 3.4), surrounding traffic (section 3.5), chemical interactions (sections 3.6 and 3.7), and VOC emitted from slurry (section 3.8). Statistical outputs are given in some cases as level of significance according to the Student-t test, histograms and correlation coefficients, but in several cases we can only argue that a correlation exists because available data only covered one event of interest (for example, one slurry application).
We have also added in the text the statistical following outputs in all Students t-test: the p-values (the probability of the null hypothesis) and the means (lines 256, 399 and in the conclusions section).

Also, whilst I understand that it is not always easy to generate perfect scientific prose, I think that the scientific English should be improved. I have listed a few items which need to be corrected below, but I suggest that the paper be re-read with spelling and grammar in mind.
Specific comments below, though this list is not exhaustive:

21: at all time(s)
Thank you for the suggestion, this was corrected for.

70 and others: nowadays – is not a very scientific term
Thank you for the suggestion, we replaced by "a well-established method" and "currently"

73: is this a quote – ""?
Yes it is indeed. It is actually from Corrsin (1975). We have added some quotes as well as the reference to Corrsin (1975).

83-98: The section reads very much like a list
The idea was to make reference to the studies reporting (to our knowledge) simultaneous NO, $NO_2$ and $O_3$ fluxes and to report their main findings. We have slightly rearranged this section with a section on forests then grasslands and other surface types

128: on the oilseed rape field
Thank you for the suggestion, this was corrected for.

131: French for December is included
Thank you for the suggestion, this was corrected for.

136: ~monthly -> approximately monthly
Thank you for the suggestion; this was corrected for (approximately once a month).

169: equation has been included in the main text
Thank you for the comment. We put the expression in a new line.

232: (Paris) – items in brackets must be elaborated upon, or not put in brackets.

Thank you for the suggestion, this was corrected for.

243: NO fluxes were slightly negative part of the time - reword
We replaced "part of the time" by "for some events"

248: and were mostly negative (deposition) – again brackets
We replaced "deposition" by "indicating deposition"

257: during bare soil periods – reword
This text was replaced by the one of the Supplementary Material (see comments of Anonymous Referee #3)

350: stats in brackets in discussion, not enough detail and no statistical output
The t-test statistical outputs are now added here and also each time it was mentioned: the p-value and the means of each sample were added.

421: numbered bullets very long – bullet points are unusual and if included, very brief
Thanks for this suggestion. We have suppressed the bullet points and add some short sentences to follow which hypothesis is addressed in each chapter.

**Answer to Anonymous Referee #2**

With the revised version, the authors have considerably improved the manuscript. However, there are still some issues that need further revision before the manuscript can be published (see detailed comments below).

**MAJOR COMMENTS**
It needs to be considered, that the Supplementary material is not an integral and equally valid part of the peer-reviewed manuscript. Therefore, information that is necessary for understanding the discussion and conclusions must not be placed in the Supplementary material but has to be included in the main manuscript. This concerns the following comments 1 and 2.

1) Supplementary section 6 should be integrated in Section 3.3.2 of the main manuscript. The latter is named 'Seasonal dynamics ...' but presently contains only one sentence (Line 260) about seasonal variations, although this is one major aspect of the presented dataset. In contrast, the overall flux statistics presented in Lines 240-250 could be shortened.
We thank anonymous Referee #2 for this remark, and we integrated the ozone deposition velocity subject in section 3.3.2.

2) Fig. S5 and the corresponding text in supplementary section 7 need to be integrated into the main manuscript, in order to make the discussion in Section 3.8.2 understandable.
We thank anonymous Referee #2 for this remark, and we integrated the ozone deposition velocity comparison subject in the main manuscript as subsection 3.3.3.

3) In Section 3.8.2 the authors should probably differentiate more between (a) empirical explanations of the observed ozone deposition velocity by measurable driving parameters like e.g. the parameterisation of Stella et al. and (b) identification of real physical/chemical adsorption and destruction processes like e.g. gas-phase chemical reactions with NO and various VOCs (above soil or within soil pore-space) or heterogeneous chemical and physico-chemical reactions at dry and wet soil surfaces.
This is an interesting comment. Actually in the three hypothesis discussed, the first one corresponds indeed to the empirical explanations based on existing knowledge of the ozone fluxes (specific to that site), while the other two hypotheses belongs to the identification of destruction processes in the gas-phase or at the surface. We however feel difficult to organise the discussion as proposed by anonymous Referee #2 since empirical models by nature include the destruction processes at the surface. We have therefore left this section as such.

4) Line 505ff.: Instead of assessing the significance of agriculture for total NOx emissions from this single site and crop, it would be much more useful to compare the observed NO emission factor to comparable emission factors used in national or international inventories (e.g. EMEP, IPCC, etc.).
Thank you for the comment. We have added the value of emission factor that is cited in the last EMEP/EEA air pollutant emission inventory guidebook, which refers to the 2006 IPCC report. Nevertheless, this is an average

value that does not take into account site-specific parameters as soil pH and fertilizer type. We have added the following sentences in the text:

"… emission factor of 0.27%, which is similar to values reported earlier for the same site (Laville et al., 2011) but one order of magnitude larger than the EMEP/IPCC default value of 0.04 (EEA, 2016). Nevertheless, this is an average value calculated with the Tier 1 approach, which does not take into account correction factors depending on soil pH or fertilizer type. This more detailed approach, the Tier 2, has not been developed for NO."

5) Line 508: The value ~5.6 t N-NO ha-1 is clearly erroneous (concerning value and/or units ha-1). 0.27% of 80 kg N ha-1 yr-1 multiplied by 26 Mha results in an annual total of 5600 t N-NO for the entire country.
Thank you for the remark, indeed the units were wrong (kt instead of t ha-1), this was corrected for. As a consequence, this means also that our estimation would be a factor 40 smaller than the emission from transports, industry and heating.

6) Line 508-512: The argumentation is contradicting here. First it is said that the annual total of agricultural NO emissions is "several hundreds of thousand" times smaller than the other emissions; and then it is argued that if concentrated within a few weeks in spring and autumn, they can nevertheless be important. A concentration of the annual emission to a few weeks (say 2-3 weeks) increases the relative magnitude (compared to other emissions during these weeks) only by a factor of 20, which is still much smaller than a factor of "several hundreds of thousand"
Thank you for this comment. We agree that during most of the year the weight of agricultural emissions probably remains small compared with other sources. Nevertheless, as pointed out in the previous comment we made a mistake in the units used in our national estimation. It turns out that considering few weeks only, would increase the relative magnitude by a factor of 20 leading to a contribution from crops only half of that from transport. This means indeed that agricultural emissions, concentrated in a short period of the year and under certain meteorological conditions, can potentially make atmospheric $NO_x$ concentrations increase above the thresholds that are critical for human health and ecosystems.

7) Figure 5b: It would be most interesting to see the effect of the chemical correction on the NO2 deposition velocity.
This is an interesting suggestion. Indeed we see that the $NO_2$ deposition velocity switches from slightly negative to slightly positive. The Figure 7b was modified and some text was added in section 3.3.4: ". In terms of deposition velocity, the ozone deposition velocity followed a clear diurnal cycle with a maximum during the day and a minimum at night. The measured $NO_2$ deposition velocity showed slightly negative values, but slightly positive ones when corrected for reactions with NO and $O_3$."

**MINOR COMMENTS AND LANGUAGE CORRECTIONS**

- Line 15 (abstract): Replace "Mean NO emissions" by "Cumulated NO emissions"
Thank you for the suggestion, this was corrected for.

- Line 19/20: This statement about the ozone deposition velocity is incomplete. It 'was significantly larger' than ...?
It was meant "than before fertilization". The expression was replaced by "deposition velocity increased significantly after organic fertilisation"

- Line 84: write more precisely "...none over an arable crop."
Thank you for the suggestion, this was corrected for.

- Line 85: This sentence needs rephrasing. It is not the question whether the gases are interacting (they do so according to general chemical and physical laws). Better write "...whether the reactions between NO, NO2 and O3 significantly influence their fluxes above crops and ..."
Thank you for the suggestion, we rephrased as you suggested.

- Line 99: It is not clear what 'adapted' means here. Better use e.g. 'suitable' or 'adaptable'.
Thank you for the suggestion, this was corrected for.

- The numbering of the (Sub-) Sections in Chapter 3 is obviously not correct. E.g. there is no Section 3.4 or 3.5
 Thank you for the suggestion, we re-numbered and ordered the sub-sections titles.

- Line 276-278: This argument should be related to the chemical correction and derivation of the surface fluxes in Section 3.8.1 and Fig. 8 later in the manuscript. Does the chemical correction of the NO fluxes lead to diurnal cycle that is more in phase with temperature?
Thank you for the interesting comment. Indeed, the derivation of surface flux lead to a diurnal cycle that is peaked later in the day, more in phase with temperature. We added comments and figure 9 to show this finding.

[Figure]

Figure 9. Diurnal cycles of ground temperature, NO flux at measurement height and at surface by the logarithmic profile.

- Line 282-284: These statements are contradicting: 0.09 nmol m-2 s-1 is not smaller than the previous result of 0.07 nmol m-2 s-1!
Thank you for the comment, we replaced "smaller" by "in the range of"

- Line 287: Correct to "...flux distributions ..."
Thank you for the comment, but we keep here the singular as we refer to one singular distribution (the NO one)

- Line 288: Change to "...the ones for the whole period"
Thank you for the comment, we add "the one for…" but we keep the singular as we refer to one singular distribution (see previous comment)

- Line 295: "Following the slurry application" is quite unspecific. Please specify the length of the time period attributed to this event.
We added here the period (two weeks)

- Line 297: What is the difference of this emission factor (0.27%) to the one in Line 295 (0.25%)? Maybe Lines 295-296 are obsolete?
Thank you for the comment. The first factor (0.27%) is referred to the whole period, while the smaller one (0.24% indeed) is referred to the nitrogen losses during two weeks after fertilization. We have synthetized the information.

- Line 309: " the soil was only humidified ... and occurred ..." This formulation is syntactically incorrect and needs rephrasing.
Thank you for the comment. We rephrased as follows:
"… while in August no significant rain event occurred after the first week. In this period indeed, the soil was only humidified by the organic manure supply (on a layer 4.8 mm thick)  that was applied on a dry soil."

- Line 359-360: It is quite unusual to present correlation coefficients in %. In addition it is not fully clear whether the normal correlation coefficient or the squared correlation is meant here. here the normal correlation coefficient (that assumes the value of 100% or 1, when the variables are linearly correlated). We added "normal" in the text. - Line 387f.: This statement needs rephrasing. The gaseous transfer in the soil is always driven by molecular diffusion. Here, the important assumption is that the soil surface deposition is quantitatively limited by molecular diffusion in the soil pores. I thus suggest to modify to: "...if molecular diffusion in the soil pores is the main limitation factor"

Thank you for the comment, we rephrased as suggested.

- Line 394: Replace "to the mast" by "to the EC measurement height"

Thank you for the comment, this was corrected for.

- LIne 408-410: This sentence can be omitted because it is a repetition of statements that have been introduced before.

Thank you for the comment, but we think that it is important here to synthetize the connection between Damköhler number and NO fluxes, also to introduce fig. 8.

- Line 418: "O3" is misplaced here.

Thanks for that comment. Yes indeed, "O3" was on the wrong side of the brackets. We corrected that typo.

- Line 427: Correct to "using the Stella et al..."

Thank you for the comment, this was corrected for.

**Answer to Anonymous Referee #3**

The manuscript titled 'Nitrogen oxides and ozone fluxes for an oilseed-rape management: influence of organic fertilisation' by Vuolo et al. discusses fluxes of NO, NO2 and O3 measured using the eddy covariance method. The paper is well written and structured and describes an interesting dataset which I believe the flux community will like to see. I recommend that the paper should be published subject to some small edits which I leave to the editors and author's discretion. I hope that my comments aid the authors and look forward to seeing the paper published.

**Comments:**

It may be prudent to change the title to 'Nitrogen oxides and ozone fluxes from an oilseed-rape management cycle: the influence of cattle slurry application'. The term organic fertilisation covers a wide variety of possibilities and as the authors point out, much of the chemistry occurring can be dominated by VOC emissions which will vary widely depending on fertiliser type and consistency.

We have followed the referee suggestion and changed the title to: "Nitrogen oxides and ozone fluxes from an oilseed-rape management cycle: the influence of cattle slurry application"

As mentioned in the previous review of the paper, it is a fairly long submission. The author's attempts to shorten the paper by converting some of the methodology section to supplementary material do help with this. I don't believe that shortening the paper further would improve its readability and would only serve to damage its scientific value.

We agree with the referee comments.

The soil pH is relatively high (7.6). Is this normal for the field or due to recent liming? There is little mention of this on nitrification rates.

The site is naturally characterized by alkaline pH that revealed to be of the same order of magnitude or larger in previous studies: 8.3 in Laville et al., 2009 and 2013 and 7.6 in Loubet et al., 2011. The following text has been added in the manuscript:

*"The soil organic carbon content was ~20 g C kg$^{-1}$, pH (in water) = 7.6, and bulk soil density was 1.3 g m$^{-3}$, in agreement with previous measurements on the same site (Laville et al., 2009 and 2011, Loubet et al., 2011). High pH are common in soils over calcareous layers and with high fine fraction content (clay and silt) as is the one of the Grignon site. It is known that alkalinity fosters the nitrification process and this range of pH is optimum for it to occur (e.g. Nieder and Benbi, 2008)."*

There is mention of the FIDES footprint analysis which shows that the effect of pollution from the cars would be minimal, but no graphical representation of this. Would it be possible to include a rough sketch of the field site and location of the roads with a representation of the footprint contribution during the measurement period?
This is a good suggestion. We have added a small map of the field with the surrounding roads.

What percentage of eddy covariance measurements passed QC steps for each compound? What is the total time coverage for each? Which version of eddy pro was used? Were any other settings changed in eddypro outside of the carboEurope settings to accommodate NO, NO2, O3. (i.e. spike removal, outliers etc…) If so please include a brief summary.
$NO_x$ and $O_3$ half-hourly fluxes were filtered by the quality check test included in EddyPro (version 5), according to the 0-1-2 labelling proposed by Mauder and Foken (2006). As recommended in the framework of the CarboEurope project, we discard fluxes with quality check index value of 2. This lead to keep the 74%, 84% and 76% of the records, for NO, O3 and NO2 respectively. The total records of NO and O3 half-hourly fluxes were 11329 (from 07/08/2012 to 13/03/2013), while for NO2 they were 2257 (during the period 14/08/2012 to 30/09/2012). This information has been added in Section 3.1, whose title has been changed into "Quality check and uncertainty in NO, $NO_2$ and $O_3$ flux measurements".

L190: It would be useful to give the reader a range of Reynolds number that would be in the turbulent range for the site in brackets. i.e (xxxx to xxxx)
We added (Re>4000) to specify the turbulent range.

I would like to see a bit more detail in how the fluxes were quality controlled. What were the cut-off values for outliers and why? Was u* limited used as a cut-off? If so, what limit was chosen and why? If not, why not?
To compute yearly averages, we used as cut-off values to exclude outliers the 99.9 percentiles separately for the positive and negative parts of the fluxes distributions, as we observed that the values outside this range were isolated and do not have physical meaning. These values resulted to be ±5nmol/m²/s for NO and $NO_2$, and -60 and +10nmol/m²/s for $O_3$.
As regards u* we did not use a threshold for it but observed that very low values of u* were already discarded with the quality check classification, that included the test for well-developed turbulence (Mauder and Foken, 2006).

L213: At what point would the authors deem the eddy covariance method unusable?
Section 3.3.1: These observations seem worrying when applying eddy covariance to such reactive compounds which are constantly changing as they disperse from sources. Later in Section 3.7 an estimate of 4 to 40% of contribution to fluxes is described based solely on estimated stats. Horizontal transfer of the species being measured in and out of the storage area of the fetch and the resultant advection effect seems to be a very significant source of uncertainty and one which cannot be fully accounted for in this study even with de-spiking etc... It is understandable that no field site is perfect, but as one of the aims of this study is to asses if eddy covariance is suitable for the measurements it seems odd to accept such a large source of error as a given. Perhaps some of the more negative aspects of the methodology should be embraced as a discussion point for future studies?
Thank you for that very sound comment indeed. This is true that the site is particularly challenging for studying $NO_x$ fluxes, and we agree that some of the most challenging aspects should be retained for future studies. We added some considerations in the conclusions, and made suggestions for additional measurements:
*"Nevertheless, random uncertainty were particularly important (>20%) during morning traffic peaks due to non-stationarity of $NO_x$ and $O_3$ mixing ratios. As concerns $NO_2$, uncertainty was even higher (up to 40%) due to the indirect measurement method. We thus recommend caution in the use of the method in non-stationary conditions, and combined measurements of horizontal gradients of mixing ratios to quantify the effect of advection. Also, additional measurements of surface mixing ratios would be useful to check the reconstruction of surface fluxes that we performed by using the logarithmic-profile model of Duyzer. Finally, high $NO_2$ to NO conversion efficiency should be assured to reduce uncertainty on $NO_2$ fluxes."*

Is it possible to report a detection limit for each of the measured fluxes?
This would be possible but makes the figures less readable. For that purpose, we prefer to stick with the hourly averages

Perhaps this comment is beyond the scope of the paper and I do not expect the authors to amend the manuscript. Was short term changes in PAR (i.e. the effect of clouds) compared to the random error of Ozone fluxes? Is this not an issue when looking at fluxes over a 30 min period when UV exposure can change so dramatically over very short time periods? If fluxes were calculated over a 60 min period instead of 30 minutes are the same fluxes

and correlations observed? Would it reduce cumulative flux uncertainty at the cost of data points and the observation of diurnal patterns?

As indeed mentioned by the reviewer this is a bit beyond the scope of this manuscript. We do have the data to look at 5 minutes PAR and O3 fluxes correlations but this would require to re-process all eddy-covariance data. We prefer to leave that question out of the way for the moment but we agree that it would be interesting to look at short-term correlations between ozone fluxes and UV radiations.

Section 3.6: How were cumulative totals estimated? Linear regression between points or using the diurnal cycles to gap fill? With such consistent patterns and correlations it seems like gap filling could be modelled relatively well?

We did not perform gapfilling but averaged fluxes that passed the quality check (74%, 84% and 76% of the records for NO, $O_3$ and $NO_2$ respectively). We preferred to avoid any reconstruction of the lacking records with models (for example the one of Henault et al., 2015) because of the high variability of emissions and dependency on many factors on which measurements were not available at the required frequency (nitrate and water soil content, soil temperature). We however acknowledge that our cumulative totals may hence be biased.

Axis Text on Fig 7 has overlapped in places

**Optional:**
I don't like the phrase 'changed sign' referring to fluxes switching between emission and uptake at different levels. L23 & L 526. If possible please re-word.

Thank you for the suggestion, we adopted the expression "switched from deposition to uptake".

L21: replace 'at all times during' with 'constantly throughout'

Thank you for the suggestion, this was corrected for.

L 37: replace 'increasing risks for' with 'exposure to which increases risk to'

Thank you for the suggestion, this was corrected for.

L48: replace 'mostly due to' with 'primarily the by-products of'

Thank you for the suggestion, this was corrected for.

L136: replace ~monthly with 'approximately once a month'

Thank you for the suggestion, this was corrected for.

L193*This is assumed to be "white noise" and……

Thank you for the suggestion, this was corrected for.

L241: replace 'on' with 'over'

Thank you for the suggestion, this was corrected for.

L246: replace 'strongest' with 'highest'

Thank you for the suggestion, this was corrected for.

L258: …it can be deduced that deposition velocities were around…

Thank you for the suggestion, this was corrected for.

L337: probably similar to those measured in September

Thank you for the suggestion, this was corrected for.

**Technical:**
Indents throughout the manuscript are inconsistent. Editorial team will correct?

Thank you for the remark, we added indents at each paragraph.

Dates are presented inconsistently throughout. Choose either (18th of February) or (18/02/16) format and stick with it.

Thank you for the suggestion, we adopt the notation dd/mm/yyyyy.

L131: *December
Thank you for the suggestion, this was corrected for.

L124 * performed on
Thank you for the suggestion, this was corrected for.

L223: *7 month period
Thank you for the suggestion, this was corrected for.

L250: replace 'than' with 'of'
Thanks for the suggestion. Changed.

L297: 0.27 or 0.25? See line295
The correct value is 0.24, we replaced it.

L330*24th
We replaced by 24/08 to keep the notation dd/mm/yyyy

L397: delete 'of'
Thank you for the suggestion, this was corrected for.